



# A Geographically Weighted Gaussian Process Regression Emulator of the GCHP 13.0.0 Global Air Quality Model

Anthony Y. H. Wong[1], Sebastian D. Eastham[2], Erwan Monier[3], Noelle E. Selin[1,4,5]

[1]Center for Sustainability Science and Strategy, Massachusetts Institute of Technology, Cambridge, MA, USA
[2]Brahmal Vasudevan Institute for Sustainable Aviation, Department of Aeronautics, Faculty of Engineering, Imperial College London, South Kensington, London, UK
[3]Department of Land, Air and Water Resources, University of California, Davis, CA, USA
[4]Department of Earth, Atmospheric and Planetary Science, Massachusetts Institute of Technology, Cambridge, MA, USA
[5]Institute for Data, Systems and Society, Massachusetts Institute of Technology, Cambridge, MA, USA

*Correspondence to*: Anthony Y. H. Wong (ayhwong@mit.edu) and Noelle E. Selin (selin@mit.edu)

**Abstract.** Air quality modelling has been an essential tool to study the impacts of socio-economic changes and policies on air quality and associated social costs due to human health impacts. However, high computational and human resource demands limit the use of state-of-the-art air quality models outside of the atmospheric science community. We address this limitation

by training Geographically Weighted Gaussian Process Regressors (GW-GPR) on the outputs of a series of perturbation experiments from the high-fidelity GEOS-Chem High Performance global chemical transport model (GCHP 13.0.0). The Gaussian Process Regressor relates changes in annual mean surface anthropogenic $PM_{2.5}$ to changes in short-lived air pollutant emissions and atmospheric $CH_4$ and $CO_2$ levels for each GCHP model grid cell. In comparison to existing widely adopted linearized and regionalized approaches, our method can account for sub-regional changes in air pollutant emission patterns

and incorporates the non-linear response of secondary air pollutants to precursor and greenhouse gas emissions. We evaluate and demonstrate the utility of our model by predicting the global distribution of $PM_{2.5}$ in 2050 (relative to 2014) under 4 sets of climate and air pollution control policy scenarios. The emulator reproduces grid cell-level changes in anthropogenic $PM_{2.5}$ ($R^2 = 0.94 – 0.99$ over the 4 scenarios tested), and associated global changes in premature mortalities at 95% confidence level, while requiring < 10 seconds of CPU time (vs. ~3000 CPU hours for GCHP) for each scenario. The emulator is also able to

capture projected global trends of population-weighted $PM_{2.5}$ from the AerChemMIP ensemble within the ensemble range. To our knowledge, the GW-GPR emulator is the first global-scale emulator operating at grid cell level with explicit consideration of non-linearities in atmospheric chemistry, climate change, and uncertainties resulting from both chemistry and climate variability. The accuracy, speed and simplicity of the emulator also show the capability of machine learning algorithms in emulating global air quality models, and make air quality modelling accessible for global climate/air pollution scenario analysis

and integrated assessment.



## 1 Introduction

Fine (aerodynamic diameter ≤ 2.5 μm) particulate matter (PM$_{2.5}$) is among the most important air pollutants at global level, threatening both human and ecosystem health. Globally, PM$_{2.5}$ exposure was estimated to be responsible for ~4 million deaths in 2019 (Sang et al., 2022), and addressing health and environmental impacts from ambient air pollution has been explicitly stated as part of the Sustainable Development Goals (United Nations, 2015). The conventional approach to evaluate the impacts of socio-economic changes and policy interventions on air quality involves producing the projected air pollutant emission inventories (and meteorological fields if direct impacts of climate change are considered) and feeding them as inputs to a chemical transport model to simulate the impacts on air pollutant concentration. This process is highly demanding in terms of human and computational resources, which limits its usage for policy analysis and integrated assessments.

To increase the accessibility of air quality modelling for the broader scientific and stakeholder communities, strategies have been developed to reduce the complexity of air quality modelling by drawing from full chemical transport model experiments, resulting in various reduced-form air quality models that are faster and easier to run while retaining a reasonable level of accuracy. One approach involves dividing the world into regions. By assuming a linear relationship between air pollutant emissions in one region (source) and the air pollutant concentrations in other regions (receptor), source-receptor (SR) matrices are constructed by running a series of chemical transport model experiments with emission perturbed individually at each region. The SR matrices can then be used as a linearized global air quality model. This approach is also useful in spatial attribution of air pollution, which is applied in the Task Force on Hemispheric Transport of Air Pollutants (HTAP) (Galmarini et al., 2017; Liang et al., 2018). Designed to be a useful tool for science-policy analysis, the TM5-FASST model (Van Dingenen et al., 2018) computes SR matrices for 56 regions of the world, and subsequently process the output into public health, agriculture and climate impact metrics. Another approach involves using output of full complexity chemical transport model to parameterize some physical and chemical processes, resulting in a reduced-order chemical transport model that can be run faster and in higher resolution (Tessum et al., 2017; Thakrar et al., 2022). These SR (Huang et al., 2023; Reis et al., 2022) and reduced-order (Camilleri et al., 2023) models have been frequently applied in recent science and policy studies, showing the utility and demand for these outputs.

However, both SR and reduced-order models rely on several simplifying assumptions, which do not always hold. Many methods rely on linearizing the relationship between emissions and concentrations, which has been shown to be a reasonable approximation when the emission change is relatively small (Van Dingenen et al., 2018), but the formation rate of major secondary air pollutants such as inorganic PM$_{2.5}$ (Ansari and Pandis, 1998) are known to respond non-linearly to precursor emissions, especially when there are shifts in chemical regimes. This issue can be relevant when exploring a wide range of climate and air quality scenarios given the large range of possible air pollutant emissions and the discrepancy in the rates of change of different precursor emissions (NO$_x$ vs NH$_3$ vs SO$_2$ for PM$_{2.5}$) (Atkinson et al., 2022; Rao et al., 2017; Turnock et al., 2020). Existing SR matrices and reduced-order models also often ignore the direct effects of climate change on air pollution





(Jacob and Winner, 2009). Garcia-Menendez et al. (2015) find that under a high-warming scenario, climate change alone can

increase population-weighted annual average $PM_{2.5}$ by 1.5 µg m$^{-3}$ between 2000 and 2100 over contiguous United States.

Recent innovations in regional reduced-form air quality models have moved beyond simple linear scaling, by applying non-linear regression techniques. Conibear et al., (2022) and Vander Hoorn et al. (2022) successfully trained Gaussian Process Regression to emulate the grid cell-level response of annual mean $PM_{2.5}$ to a large range (-100% to +50%) of sectoral emission perturbations over China and the Perth greater metropolitan region respectively, with regional chemical transport model

perturbation experiments as training data. Colette et al. (2022) applied multivariate quadratic regressions to emulate the simulated $PM_{2.5}$ response to emission control policies over Europe, achieving an accuracy of <2% relative error in 95% of grid cells. Meanwhile, a geographically-weighted linear regression emulator was shown to reproduce $PM_{2.5}$ response to precursor emissions from the parent chemical transport model within 10% accuracy over Europe (Pisoni et al., 2017).

Building on these regional scale applications, we combine Geographic Weighting and Gaussian Process regression (GW-GPR)

techniques to emulate the output of a high-fidelity global chemical transport model, GEOS-Chem High Performance (GCHP) driven by meteorological data from multiple climate simulations with the Community Atmosphere Model (CAM, collectively GCHP-CAM) (Eastham et al., 2023) This results in a global reduced-form air quality model that can account for spatially heterogenous pollutant emission changes and non-linearity in atmospheric chemistry under multiple climate scenarios, and provide robust uncertainty estimates, without drastically increasing the computational cost.

**2 Method**

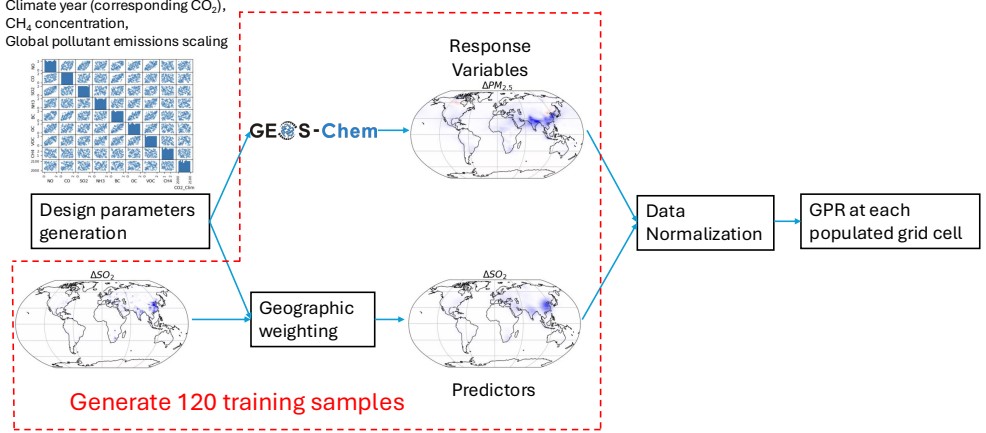

**Fig. 1 Schematic showing the steps involved in the development of the emulator**

We emulate the response of annual mean surface $PM_{2.5}$ to changes in air pollutant emissions and climate modelled by a high-fidelity global chemical transport model, GEOS-Chem High Performance (GCHP). Here, we provide descriptions for our

GCHP model setup and experiments, and the process of constructing the emulator (Fig. 1).



## 2.1 GEOS-Chem High Performance Model

We use the GCHP-CAM modelling system described and evaluated in Eastham et al. (2023). The modelling system is based on a customized version of GCHP 13.0.0 (The International GEOS-Chem User Community, 2024) that can be driven by the modelled meteorological fields of the Community Atmosphere Model (CAM) version 3.1. Here we provide a brief description
of the modelling system and specific setups for our work.

GCHP (Eastham et al., 2018) simulates $PM_{2.5}$ by resolving the chemistry, transport, emission and deposition of relevant chemical species. Oxidant chemistry is simulated using a coupled $VOC$-$CO$-$NO_x$-$O_3$-aerosol-halogen chemical mechanism (Sherwen et al., 2016). $PM_{2.5}$ includes contribution from nitrate, sulphate, ammonium, black carbon (BC), organic carbon (OC), fine dust, sea salt and secondary organic aerosols (SOA). The formation of secondary inorganic aerosols is simulated
by considering the thermodynamic equilibrium of the $NH_4^+$–$Na^+$–$SO_4^{2-}$–$NO_3^-$–$Cl^-$–$H_2O$ system through ISORROPIA II (Fountoukis and Nenes, 2007). Organic aerosols are assumed to be non-volatile.

The model is driven with climate projections simulated by the IGSM-CAM (Monier et al., 2013) a modelling framework that links the MIT Integrated Global System Model (IGSM, Monier et al., 2018) to the National Center for Atmospheric Research (NCAR) Community Atmosphere Model (CAM) 3.1 (Collins et al., 2006). The simulations are described in detail in Monier
et al. (2015). The global climate model is run from 2000 – 2100 with a horizontal resolution of 2° × 2.5° on 26 vertical layers up to 2.2 hPa. We choose the high-warming "REF" scenario (10 W/m$^2$ in 2100, resulting in 4.3 °C warming in 2080 – 2100 versus 1990 – 2009) to provide samples across a wide range of warming and $CO_2$ concentration. The meteorological data from this climate projection is processed into the format of the Modern Era Retrospective for Research and Analysis version 2 (MERRA-2) (Gelaro et al., 2017) meteorological fields used by native GCHP. GCHP is run at C48 (~200km) horizontal
resolution with the same vertical layers with the CAM simulations. The model output is remapped into a 2° latitude × 2.5° longitude horizontal grid conservatively (Jones, 1999).

Anthropogenic emissions of non-greenhouse gas (GHG) air pollutants are from the Community Emission Data System (Hoesly et al., 2018). Biogenic volatile organic compounds (BVOC) emissions follow Guenther et al. (2012) with isoprene inhibition by $CO_2$ (Possell and Hewitt, 2011; Tai et al., 2013) included. Soil $NO_x$ emissions follows Hudman et al. (2012). While BVOC
and soil $NO_x$ emissions are both calculated online (and therefore respond to climate and atmospheric $CO_2$ level), mineral dust (Meng et al., 2021) and lightning $NO_x$ (Murray et al., 2012) emissions are held at 2014 level. The monthly surface $CH_4$ concentration is prescribed by spatially kriging the observations from National Oceanic and Atmospheric Administration Global Monitoring Laboratory Cooperative Air Sampling Network at 2014 level. Scaling of anthropogenic emissions and atmospheric $CH_4$ concentration in the training sets are described in the next section.

Aerosol concentrations are archived at daily time resolution, and subsequently processed into annual mean surface total anthropogenic $PM_{2.5}$. Total $PM_{2.5}$ mass is calculated from the aerosol concentration by considering aerosol hygroscopic growth at 35% relative humidity, aligning with the $PM_{2.5}$ measurement standard of the United States Environmental Protection Agency (USEPA) (Latimer and Martin, 2019). Anthropogenic $PM_{2.5}$ mass is calculated by the above method, but only summing a



subset of aerosol species (sulphate, nitrate, ammonium, BC and OC) while leaving out the aerosol species that are mostly from
natural sources (dust, sea salt).

## 2.2 GCHP-CAM experiments

### 2.2.1 Generating the training set through perturbation experiments

To effectively sample the sensitivity of $PM_{2.5}$ over a wide range of climate and air pollution emissions, we generate the training
set from a series of GCHP-CAM perturbation experiments by manipulating 9 input variables that affect $PM_{2.5}$ and oxidant
levels: 7 air pollutant emissions ($NO_x$, $SO_2$, $NH_3$, NMVOC, BC, OC, carbon monoxide (CO)) that are commonly provided by
integrated assessment models (Gidden et al., 2019), $CH_4$ concentration, and global warming level. Global warming level cannot
be directly implemented in GCHP as a scaling factor; the global warming scaling is implemented by driving the model with
simulated meteorological fields at the year with the closest $CO_2$ level under the "REF" scenario.

sets of scaling factors (0 to 1) for the 9 input variables are generated following a Latin Hypercube Sampling (LHS) (McKay
et al., 1979) strategy. However, the changes in emissions of different pollutants are correlated, since different air pollutants
often share similar emission sources (e.g. combustion). To account for co-emissions of air pollutants, we calculate the spatial
correlations of the emissions of the 7 air pollutants from CEDS between 2000 and 2017. The correlation of $CH_4$ and global
warming level with other variables are set to be 0 to provide independence between climate and air pollution control policies.
This correlation matrix is then used to rearrange the LHS scaling factors using an Iman-Conover Transform (Conover and
Iman, 1982).

We first run GCHP-CAM with meteorological fields from 1st Oct 2013 to 31st Dec 2014 and 2014 anthropogenic emissions,
$CH_4$ and $CO_2$ concentration under the "REF" scenario to generate the baseline for comparison, with the first 3 months of model
run discarded as spin-up (output not used). The rearranged LHS scaling factors sets are then linearly mapped to the range of
inputs (Table 1), and each set corresponds to a 1-year perturbation simulation, again with an extra 3 months before as spin-up.
Each of the perturbation simulations are driven by the globally scaled 2014 anthropogenic air pollutant emissions and surface
$CH_4$ levels. The scaling factor for $CH_4$ (0.5 to 2.5) is chosen to enclose the range of $CH_4$ concentration in 2100 projected by
Meinshausen et al. (2020) over all scenarios. Instead of total radiative forcing, the global warming level is parameterized as
atmospheric $CO_2$ level as $CO_2$ is projected to dominate (68 – 85%) total radiative forcing in the 21st century (Meinshausen et
al., 2020). In addition, atmospheric $CO_2$ level also directly affects isoprene emission, which could affect atmospheric oxidant
(e.g. OH, $O_3$) (e.g. Tai et al., 2013), and therefore potentially secondary inorganic aerosol formation.

| Variables | Range |
|---|---|
| Air pollutant emission scaling factor | 0 – 2 |
| Surface $CH_4$ concentration scaling factor | 0.5 – 2.5 |
| Atmospheric $CO_2$ level | 369.9 – 813.5 ppm |

**Table 1. Range of scaling factors and $CO_2$ concentration of the perturbation experiments.**



### 2.2.1 IGSM-GAINS-TAPS combined air quality and climate legislation scenarios

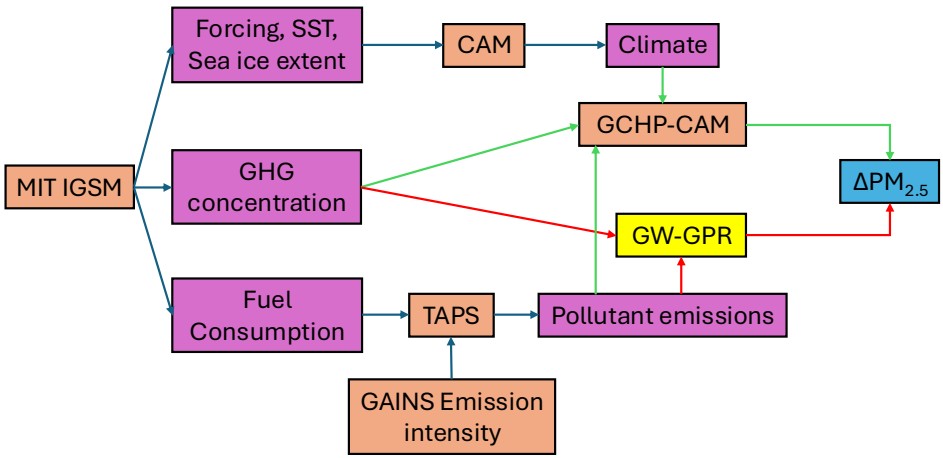

**Fig. 2 Schematic of GW-GPR model evaluation using IGSM-GAIN-TAPS climate and air pollution scenarios. Orange boxes represent existing modelling systems, purple boxes represent data sets. Yellow represents our newly developed emulator (GW-GPR). Green and red arrows represent how GCHP-CAM and GW-GPR predicts changes in PM2.5, respectively.**

To assess the utility of the emulator within the context of integrated assessment modelling, we evaluate the ability of the emulator in reproducing GCHP output anthropogenic $PM_{2.5}$ over 2 climate (Current Trend (CT) and Accelerated Action (AA))

× 2 air pollution control (Current Legislation (CLE) and Maximum Feasible Reduction (MFR)), in total 4 scenarios (CT_CLE, CT_MFR, AA_CLE, AA_MFR) in 2050. CT assumes the implementation of Nationally Determined Contributions (NDCs) from Paris Agreement through 2030. Despite such effort, climate is not stabilized, and global mean temperature continues to increase. AA assumes the extension of these initial NDCs to align with the long-term goal of Paris Agreement, therefore the ability of limit and stabilize anthropogenic warming to 1.5 °C at 2100 with at least 50% probability. The CLE scenario assumes

complying with existing region- and source-specific air pollutant emission limits, while the MFR scenario assumes increasing deployments of currently available lowest-emitting technologies.

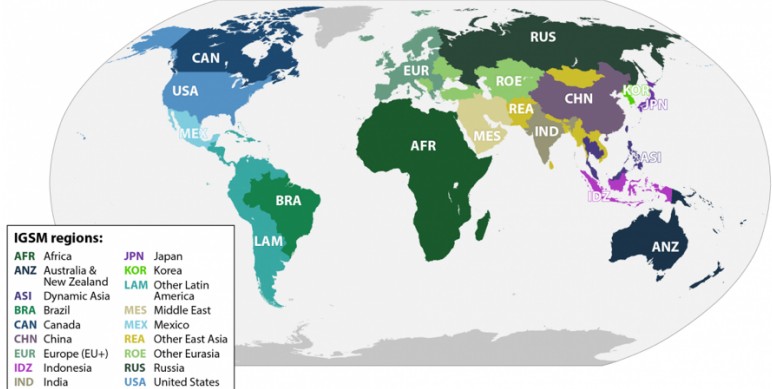

**Fig. 3 IGSM region definition (retrieved from https://globalchange.mit.edu/research/research-tools/eppa, date accessed: 31st March 2025)**





The climate scenarios are generated from the MIT IGSM framework. The human system component of IGSM, Economic Projection and Policy Analysis model version 7 (EPPA7) (Chen et al., 2022), is a global multi-sector (22 sectors) multi-region (18 regions, Fig. 3) recursive-dynamic computable general equilibrium model. As a part of climate/economic scenario projection, EPPA provides regionalized and sectorized consumptions of different fuel types, and populations. The yearly global average atmospheric GHG concentrations is then derived by driving the MIT Earth System Model (MESM) (Sokolov et al.,

2018) with corresponding EPPA output.

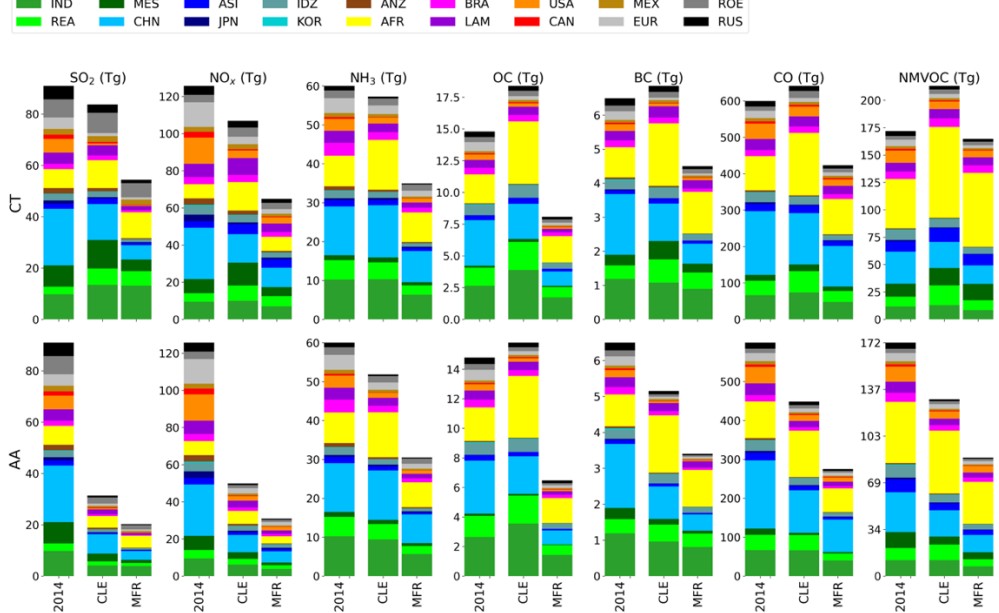

**Fig. 4 Total and regional emissions for the four IGSM-GAINS-TAPS scenarios considered in this study at base year and 2050.**

The projected trends of air pollutant emission intensities under CLE and MFR scenarios are from Greenhouse gas-Air pollution INteractions and Synergies (GAINS) (Amann et al., 2011), based on GAINS4/ECLIPSE (Evaluating the Climate and Air

Quality Impacts of Short-lived Pollutants) v6b data (GAINS Developer Team, 2021; Klimont et al., 2017; Smith et al., 2020; Stohl et al., 2015). Future air pollutant emissions for each scenarios can be derived through the Tool for Air Pollution Scenarios (TAPS) (Atkinson et al., 2022) by considering climate and air pollution policies independently. The resulting air pollutant emissions of each of the 4 combined scenarios are shown in Fig. 4.

We perform 10 years of GCHP-CAM simulations for each IGSM-GAINS-TAPS scenarios with their respective anthropogenic

emissions. The meteorological years (2031 – 2041 for AA and 2040 – 2050 for CT) are chosen to match the level of warming of our meteorological fields ("REF" scenario).

## 2.3 Geographically Weighted Gaussian Process Regression (GW-GPR) Emulator

We use Gaussian Process Regression (GPR) (Williams and Rasmussen, 1995) to relate the changes in pollutant emissions and climate with the corresponding changes in annual mean PM$_{2.5}$ level at grid cell level. GPR is a non-linear and non-parametric





regression algorithm that does not require prior assumptions of the functional relationship between input and output variables. Instead, predictions are made by assuming the training and prediction sets follow a joint multivariate normal distribution:

$$\begin{pmatrix} Y_1 \\ Y_2 \end{pmatrix} = N \begin{pmatrix} \mu_1 \\ \mu_2 \end{pmatrix}, \begin{matrix} \Sigma_{11} & \Sigma_{12} \\ \Sigma_{21} & \Sigma_{22} \end{matrix} \quad (1)$$

Where $Y_1$ is the random variable representing the prediction, and $Y_2$ is the random variable representing the response from training set. $\mu_1$ and $\mu_2$ are their means, and $\Sigma_{11}$, $\Sigma_{12}$, $\Sigma_{21}$, $\Sigma_{22}$ are the covariance matrix blocks. The prediction process can be

viewed as finding $(Y_1|Y_2 = a) \sim N(\mu', \Sigma')$, where $a$ is the model output from training set. Thus, the mean ($\mu'$) and variance ($\Sigma'$) of the prediction can be calculated as:

$$\mu' = \mu_1 + \Sigma_{12}\Sigma_{22}^{-1}(a - \mu_2) \quad (2)$$

$$\Sigma' = \Sigma_{11} - \Sigma_{12}\Sigma_{22}^{-1}\Sigma_{21} \quad (3)$$

By setting the prior mean of prediction as 0 and proper normalization during training process, $\mu_1$ and $\mu_2$ = 0. Then $\mu'$ is

essentially a sum of **a** weighted by the correlations between the training and prediction vectors ($\Sigma_{12}\Sigma_{22}^{-1}$). The elements of the correlation matrix take the form:

$$\sigma_{ij} = k(x_i, x_j) \quad (4)$$

Where $x_i$ and $x_j$ are the input vectors at each training or prediction points, and $k$ is the covariance function (kernel) that can be chosen to control the shape and smoothness of the prediction. We use a sum of anisotropic kernels to represent the nature of

our problem (smooth functions (rational quadratic function) with unknown points of chemical regime change + local interactions among variables (Matern 3/2 function) + noise from climate variability (white noise)). We use the GPR as implemented in Scikit-learn version 1.3.2 (Pedregosa et al., 2011), and only train a GPR model for each populated (population density > 1 person/km$^2$) model grid cell. As the predictions are random variables, the uncertainty and confidence interval of each prediction can be derived by its standard deviation.

We apply geographic weighting to the emission fields of individual pollutant species to emulate the process of chemical transport of emitted species. An isotropic 2D Gaussian dispersion kernel is applied to calculate the effective air pollutant emission changes ($\Delta E_{weighted,x}$) at each grid cell $x$:

$$\Delta E_{weighted,x} = \sum_y e^{-d_{y,x}^2/2L^2} \Delta E_y \quad (5)$$

Where $\Delta E_y$ is the emission change all individual grid cells considered within the dispersion kernel ($y$), $d_{y,x}$ is the distance

between grid cell $y$ and $x$, and $L$ is the dispersion length scale. The length scales (in the unit of grid cell, 1 grid cell = 2° latitude × 2.5° longitude) of each species are chosen to match their atmospheric lifetime. The dispersion kernel is implemented by the Gaussian Blurring algorithm as in Scipy version 1.10.1 (Virtanen et al., 2020). We assign $L$ = 1 grid cell for relatively short-lived species (NO$_x$, NH$_3$, NMVOC), $L$ = 2 grid cells for longer-lived species (BC, OC, SO$_2$), and $L$ = 3 grid cells for CO. We note that some previous regional studies (e.g. Pisoni et al., 2017) have treated the parameters of the dispersion kernel as

optimizable hyperparameters. However, our GCHP-CAM experiments are conducted with uniform global scaling factors for emission fields. After the variable normalization procedure, training configurations with different dispersion kernels would



effectively collapse to the same 120 sets of global scaling factors prescribed in the GCHP-CAM experiments. Therefore, our training set cannot be used to directly optimize the dispersion kernel.

While we do not perform rigorous optimization of the dispersion kernel, we explore the associated uncertainties by sensitivity

simulations of training the GW-GPR emulator with halved and doubled dispersion length scales of the GW-GPR emulator. The result is discussed in Section 3.3.2. The input variables are normalized by their corresponding global maximum value after the geographic weighting. Since the output variables are not blurred, and $\mu_2 = 0$ simplifies computation, the output variables are normalized by local mean and maximum at each grid cell.

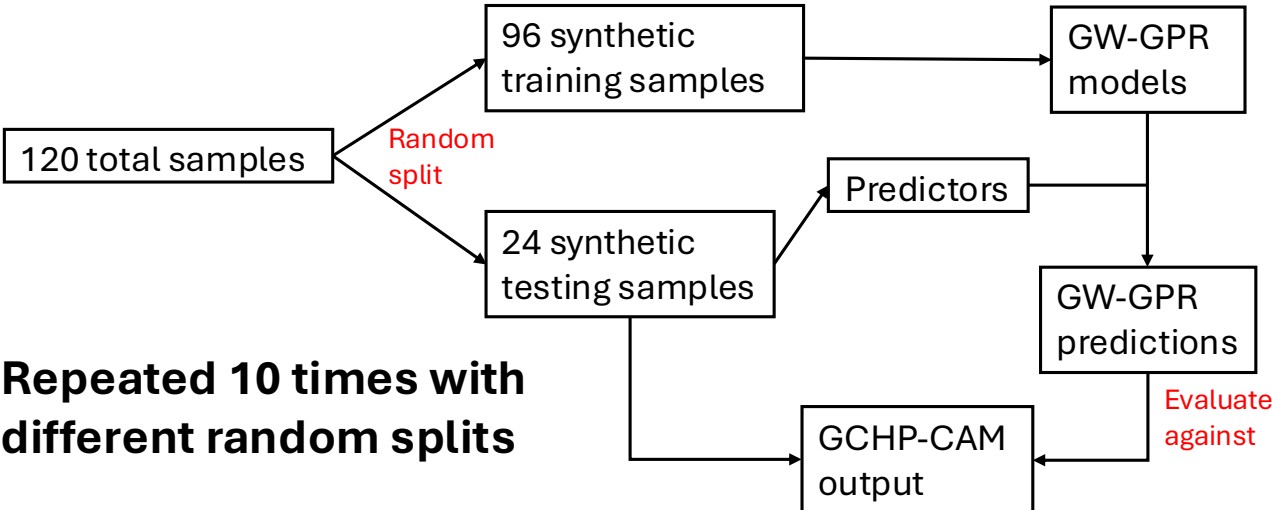

**Fig. 5 Schematic of the cross-validation procedure**

We evaluate the generalization ability of our models using the repeated random sub-sampling technique. For each repetition (Figure 5), we randomly split the data into training (80%) and testing (20%) sets. New GW-GPR models are built from the synthetic training set and predictions are mode over the synthetic testing set at grid cell level. This process is repeated 10 times, and the performance metrics are calculated using all 10 synthetic testing sets and corresponding predictions. In addition, we

perform Sobol global sensitivity analysis (Sobol′, 2001) by drawing $1024 \times (2 \times$ number of input variables $+ 2)$ samples to compute the total sensitivity indices for each input variables using Saltelli sampling (Saltelli et al., 2010) at each grid cell to using SALib 1.4.8 (Herman and Usher, 2017; Iwanaga et al., 2022), which helps us identify the importance of each input variable for different locations.





## 2.4 AerChemMIP data

| | SSP3-7.0 Ensemble members | SSP3-7.0-lowNTCF ensemble members | Model references | Data references |
|---|---|---|---|---|
| EC-Earth3 | 2 | 2 | (Döscher et al., 2022) | (Consortium (EC-Earth), 2023) |
| GFDL-ESM4 | 1 | 1 | (Dunne et al., 2020) | (Horowitz et al., 2023) |
| GISS-E2.1-G | 16 | 4 | (Kelley et al., 2020) | (NASA Goddard Institute For Space Studies (NASA/GISS), 2023a, b) |
| GISS-E2.1-H | 1 | 1 | (Kelley et al., 2020) | (NASA Goddard Institute For Space Studies (NASA/GISS), 2023c) |
| IPSL-CM5A2-INCA | 1 | 1 | (Sepulchre et al., 2020) | (Boucher et al., 2023) |
| MIROC-ES2H | 1 | 0 | (Kawamiya et al., 2020) | (Watanabe et al., 2023) |

**Table 2. The numbers of ensemble members with all 5 anthropogenic PM$_{2.5}$ components (sulphate, nitrate, ammonium, BC, OC) available for SSP3-7.0 and SSP3-7.0-lowNTCF scenarios from the AerChemMIP archive.**

To further assess and demonstrate the utility of the emulator, particularly under global change scenarios, we also compare the

output of the GW-GPR emulator with the output from the Aerosol Chemistry Intercomparison Project (AerChemMIP) under 2 Shared Socio-economic Pathways (SSP)-based scenarios: 1) the standard SSP3-7.0 "Regional Rivalry" scenario (radiative forcing = 7.0 W m$^{-2}$ at 2100); and 2) a variant of SSP3-7.0 with same socio-economic assumptions as the standard SSP3-7.0, but stronger air quality control measures, resulting in lower emissions of Near Term Climate Forcers (SSP3-7.0-lowNTCF) (Fujimori et al., 2017). AerChemMIP (Collins et al., 2017) is endorsed by the Coupled-Model Intercomparison Project 6

(CMIP 6) to quantify the impacts of aerosols and chemically reactive gases on climate. There are minimum model complexity requirements (atmosphere-ocean general circulation model with tropospheric aerosols driven by pollutant emission fluxes) to participate in AerChemMIP ensemble.

We calculate the anthropogenic PM$_{2.5}$ (sum of sulphate, nitrate, ammonium, BC and OC) from AerChemMIP archive in an identical manner as for GCHP (summing contributions from individual components after applying the hygroscopic growth

factor at 35% RH). We only include ensemble members with the output of all five anthropogenic PM$_{2.5}$ components available. The models selected and numbers of realizations included for each model are summarized in table 2.

The emulator calculates the decadal changes in anthropogenic PM$_{2.5}$ concentrations relative to 2020 over 2030 – 2090, using surface air pollutant emissions (calculated as anthropogenic + open burning emissions) and GHG concentration (Meinshausen et al., 2020) provided by the Input4MIP repository. As air pollutant emissions are provided every 10 years, one emulator

prediction is done per decade. For each ensemble member in the AerChemMIP archive, the corresponding decadal changes in





anthropogenic $PM_{2.5}$ are calculated by comparing the decadal average anthropogenic $PM_{2.5}$ with that of the first decade (2015 – 2024). The model-specific decadal changes in anthropogenic $PM_{2.5}$ is then calculated by averaging the result from all the ensemble members from the corresponding model. AerChemMIP and Input4MIP data are retrieved via the search engine of Earth System Grid Federation (ESGF) (https://aims2.llnl.gov, last access: 26th Aug 2024).

**2.5 Health Impact Calculation**

For the four climate and air pollution control scenarios, we also estimate the impacts of changes in anthropogenic $PM_{2.5}$ on public health through premature mortalities. GCHP and emulator output are upsampled from $2° × 2.5°$ to $0.5° × 0.5°$ using the nearest neighbour algorithm, which matches the horizontal resolution of the age-specific population data we use (Gridded Population of the World version 4.11, last access: 19th Apr, 2024) (Center For International Earth Science Information Network-CIESIN-Columbia University, 2018). Country-level baseline age- and cause-specific mortality rates are provided by the World Health Organization (WHO) (WHO, 2018). The age- and cause-specific changes in the annual mortality due to chronic $PM_{2.5}$ exposure for scenario $i$ ($\Delta Mort_i$) is calculated from the relative mortality risks under the baseline (total $PM_{2.5}$ from the 2014 baseline run) ($RR_{base}$) and each scenario $i$ ($RR_i$):

$$\Delta Mort_i = Mort_{base}\left(\frac{RR_i - RR_{base}}{RR_{base}}\right) \quad (6)$$

where $Mort_{base}$ is the age- and cause-specific mortalities from the WHO.

We use the age-specific non-linear Concentration Response Functions from the Global Exposure Mortality Model (Burnett et al., 2018) to calculate $RR_i$ and $RR_{base}$ for non-communicable diseases and lower respiratory infections attributable to outdoor $PM_{2.5}$ pollution. Since our emulator focuses on the changes and corresponding impacts in anthropogenic $PM_{2.5}$, the synthetic $PM_{2.5}$ level for scenario $i$ at each grid cell is calculated as $PM_{2.5,base} + \Delta PM_{2.5,i}$, where $PM_{2.5,base}$ is the modelled total $PM_{2.5}$ in 275 year 2014, and $\Delta PM_{2.5,i}$ is the modelled/emulated change in anthropogenic $PM_{2.5}$ for scenario $i$.

**3 Comparisons with GCHP-CAM**

In this section we discuss and explain the performance of our GW-GPR emulators against GCHP-CAM simulations, measured by the response of anthropogenic $PM_{2.5}$ pollution level and associated premature mortalities.

**3.1 Computing resource requirement**

GCHP-CAM requires 2400 – 3000 CPU hours (Intel Xeon Processor E5-2679A v4, processor base frequency = 2.6 GHz)) to simulate $PM_{2.5}$ for each 1-year run. The one-time operation of fitting the GW-GPR emulator requires 280 CPU hours (Intel Xeon Processor E5-2670, processor base frequency = 2.6 GHz). Once trained, the emulator requires approximately 10 CPU seconds to generate global $PM_{2.5}$ predictions for one scenario (Intel Xeon Processor Silver 4214R, processor base frequency = 2.4 GHz). This demonstrates the magnitude of the speed up offered by the emulator.





## 3.2 Emulator validation and sensitivity

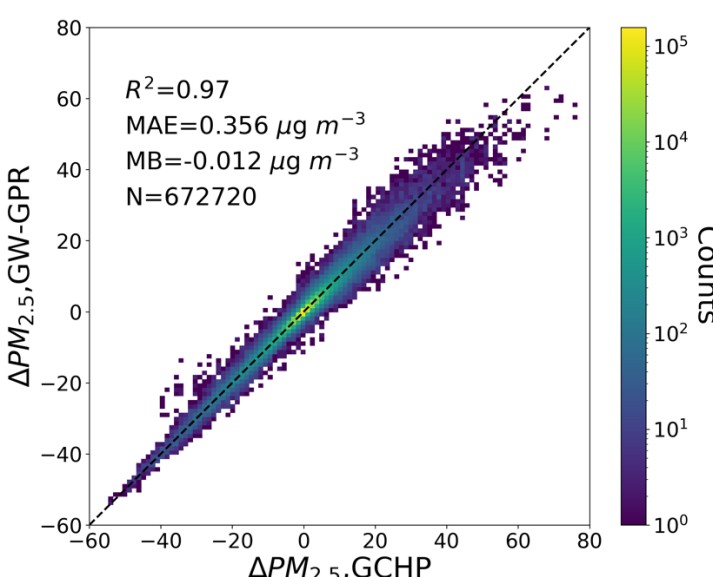

**Fig 6. 2D histogram from the grid cell by grid cell comparison between ΔPM2.5 predicted by the GW-GPR emulator (ΔPM2.5, GW-GPR) and that simulated by GCHP-CAM (ΔPM2.5, GCHP) from the 10-fold random sub-sampling cross-validation.**

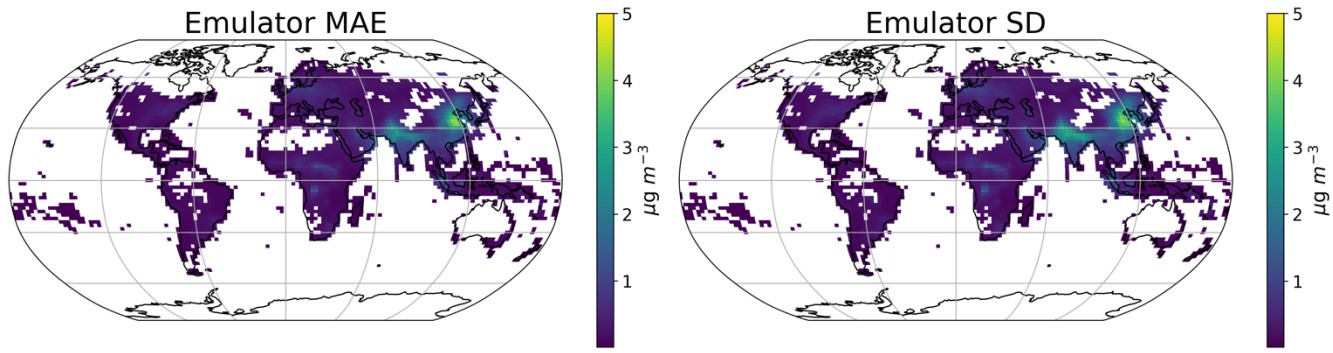

**Fig. 7 The mean absolute error (MAE) of emulator prediction against the parent model (GCHP-CAM), and the average standard deviation of emulator predictions (indicative of uncertainties from climate variability and chemistry) at grid cell level (240 data points at each grid cell)**

Fig. 6 shows the result grid cell by grid cell comparison of changes in annual mean anthropogenic PM$_{2.5}$ (ΔPM$_{2.5}$) predicted by the GW-GPR emulator against that simulated by its parent model (GCHP-CAM) across all the data points generated by the random subsampling. The GW-GPR emulator can predict ΔPM$_{2.5}$ GCHP-CAM with reasonable accuracy ($R^2$ = 0.97, mean absolute error (MAE) = 0.356 μg m$^{-3}$) and minimal overall bias (mean bias (MB) = -0.012 μg m$^{-3}$). Fig.7 shows the spatial distribution of grid cell level MAE of the GW-GPR emulator, and the emulator output standard deviation (which can characterize the uncertainty of emulator output). The largest MAE is found over Northern China and Northern India (up to 5 μg m$^{-3}$), where the anthropogenic PM$_{2.5}$ and precursor emissions are very high in the base year of 2014. The emulator output





standard deviation have similar magnitudes and spatial distributions (spatial $R^2 = 0.99$) as MAE, indicating that emulator output standard deviation is an appropriate measure of chemical and climate uncertainties of emulator predictions relative to the parent model.

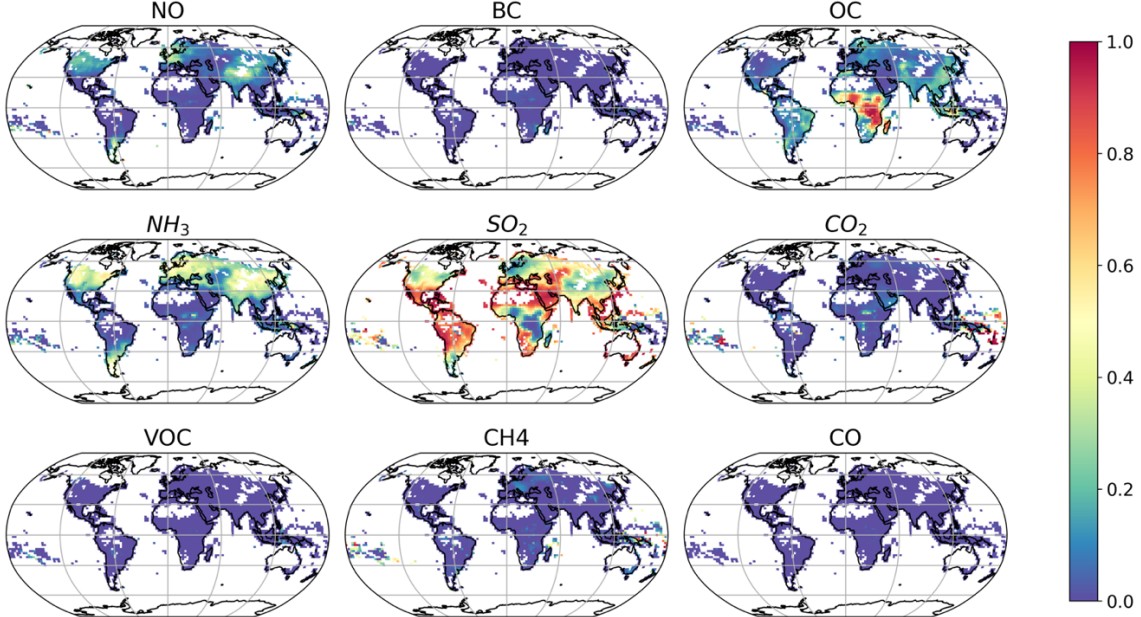

**Fig. 8 Spatial patterns of Sobol Total Sensitivity Indices for each predictor for $\Delta PM_{2.5}$.**

Fig. 8 shows the normalized Sobol Total Sensitivity Indices of the GW-GPW emulator to each of the input variables (in the unit of fractional rather than absolute changes), which measure how much each input variable is responsible for the variance in the output over the whole domain of input data, including the interaction among variables. In other words, the Sensitivity Indices indicate how important the specific input variable is in controlling the output. The importance of input variables is spatially heterogenous. Over North America, Europe, $\Delta PM_{2.5}$ is mostly sensitive to $SO_2$ (46 – 57% of total sensitivity index)

and $NH_3$ (28 – 32%), and to a lesser extent NO emissions (10 – 14%). The pattern of total sensitivity indices over India and China are similar (9 – 13% for NO, 21 – 37% for $NH_3$ and 36 – 57% for $SO_2$), but the sensitivity of $\Delta PM_{2.5}$ to OC (13% vs 3 – 7% over North America and Europe) is higher over these regions. For most of the rest of the northern hemisphere, $\Delta PM_{2.5}$ is primarily sensitive to $SO_2$ emissions (e.g. >80% over Mexico and Middle East). Over the southern hemisphere, $\Delta PM_{2.5}$ remains highly sensitive to $SO_2$ emissions (47% over Indonesia – 76% over Brazil). In Brazil, Indonesia, and Africa, $\Delta PM_{2.5}$ is also

sensitive to OC emissions (15% - 38%).  Sensitivity of $\Delta PM_{2.5}$ to BC is relatively low globally (mean = 0.006 over the globe). This is because BC is largely co-emitted with OC, while the OC emissions are always around 1 – 2 times larger than BC emission by mass. Thus, the variance attributable to BC is mostly captured by the variance attributable to OC. The sensitivity index of $CO_2$, $CH_4$, VOC and CO are also relatively low (<3%) globally, except over the certain regions with low anthropogenic





emissions (tropical Pacific islands, edge of the Amazon and central African rainforests), reflecting the fact that our emulator

does not account for secondary organic aerosols.

### 3.3 Emulator performance over IGSM-GAINS-TAPS scenarios

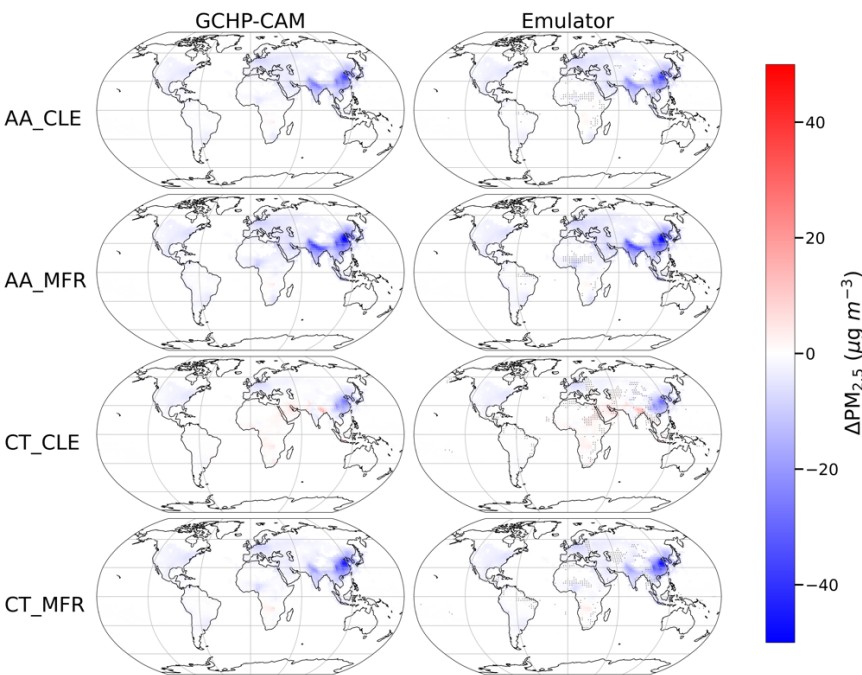

**Fig. 9 Spatial patterns of GCHP-CAM and emulator predicted ΔPM$_{2.5}$ for each of the 4 IGSM-GAINS-TAPS scenarios at 2050 (relative to 2014). Only results in grid cells with population density > 1 person km$^{-2}$ are shown. The dots show where GCHP-CAM**
**output does not fall within the 95% confidence interval of emulator prediction.**

| Scenario | Regression technique | $R^2$ | MAE (μg m$^{-3}$) | MB (μg m$^{-3}$) | % of grid cells agreeing within 1 (2) SD |
|----------|---------|------|------|-------|------|
| AA_CLE | GRP | 0.99 | 0.25 | +0.04 | 76.2 (94.3) |
|        | MLR | 0.98 | 0.35 | +0.02 | |
| AA_MFR | GPR | 1.00 | 0.20 | +0.05 | 84.8 (96.0) |
|        | MLR | 0.99 | 0.29 | -0.08 | |
| CT_CLE | GPR | 0.94 | 0.42 | +0.10 | 58.7 (82.9) |
|        | MLR | 0.93 | 0.47 | +0.04 | |
| CT_MFR | GPR | 0.98 | 0.34 | +0.07 | 68.3 (87.6) |
|        | MLR | 0.98 | 0.39 | -0.01 | |

**Table 3. Gaussian Process Regression (GPR) and multilinear regression (MLR) emulator performance metrics (spatial coefficient of determination ($R^2$), mean absolute error (MAE), mean bias (MB), computed at grid cell level, N = 2803) for each IGSM-GAINS-TAPS scenarios, relative to GCHP-CAM output. The rightmost column indicates the percentage of gridc ells that the GPR emulator prediction agrees with GCHP-CAM output within 1 (2) standard deviation (prediction uncertainty of emulator).**





Fig. 9 shows the GCHP-CAM and GW-GPR (emulator) output $\Delta PM_{2.5}$ (2045 – 2054 mean vs 2014) over each IGSM-GAINS-TAPS scenario at grid cell level. The global performance metrics are shown in table 3. Generally, the emulator performs comparably to that in the random subsampling evaluation ($R^2$ = 0.94 – 0.99, MAE = 0.20 – 0.42 µg m$^{-3}$). 58.7 – 84.8% and 82.9% (96%) of the grid cells have emulator predictions agreeing with GCHP-CAM within 1 (2) standard deviation of emulator output respectively (computed with eq. 3).

When the predicted spatial distributions of $\Delta PM_{2.5}$ are converted into premature mortalities using the GEMM CRF, we find that the GCHP-CAM and emulator output produce similar impacts on global premature mortalities over the 4 IGSM-GAINS-TAPS scenarios tested (differences within 1.2%) that agree within the range of uncertainty due to the GEMM CRF parameters (Fig. 10). This shows the emulator's ability to reproduce both the magnitudes and spatial distributions of $\Delta PM_{2.5}$ from GCHP-CAM, and the suitability of emulator output for public health impact calculation at global level.

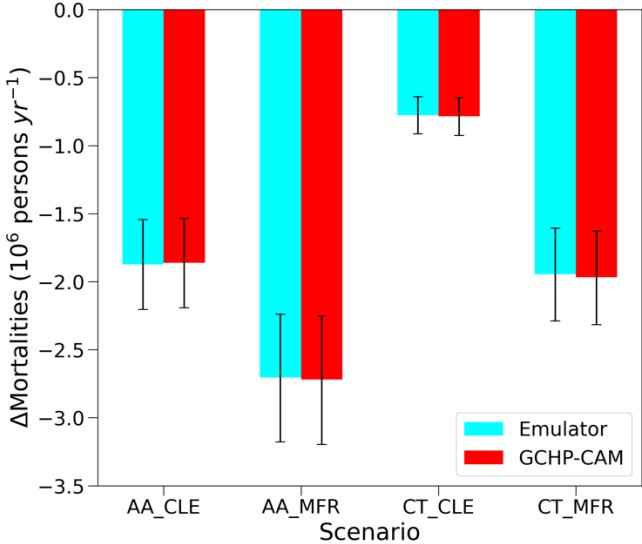


**Fig. 10 Changes in global annual premature mortality attributable to PM$_{2.5}$ exposure under each of the four scenarios between 2050 and 2014, calculated from the emulator and GCHP-Chem output $\Delta$PM$_{2.5}$. The error bars represent the uncertainties due to GEMM CRF parameters, calculating by applying the 2.5 and 97.5 percentile estimate of the GEMM CRF parameters.**

In general, the emulator performs the best over the western hemisphere (longitude < -20°), where the emulator error is within

2 µg m$^{-3}$ (MAE = 0.1 µg m$^{-3}$), and 77.4% of emulator predictions agrees with GCHP-CAM output within 1 emulator output standard deviation. In contrast, the emulator output shows consistent high bias of up to 2 µg m$^{-3}$ over the 4 scenarios over the Sahel. Around the Bay of Bengal, emulator output does not agree with GCHP-CAM output with 1 emulator output standard deviation in a large portion of grid cells. In the subsections below, we will explore the potential sources of error by comparing the result presented above with that from alternative emulator architectures.





### 3.3.1 Comparison with linear model

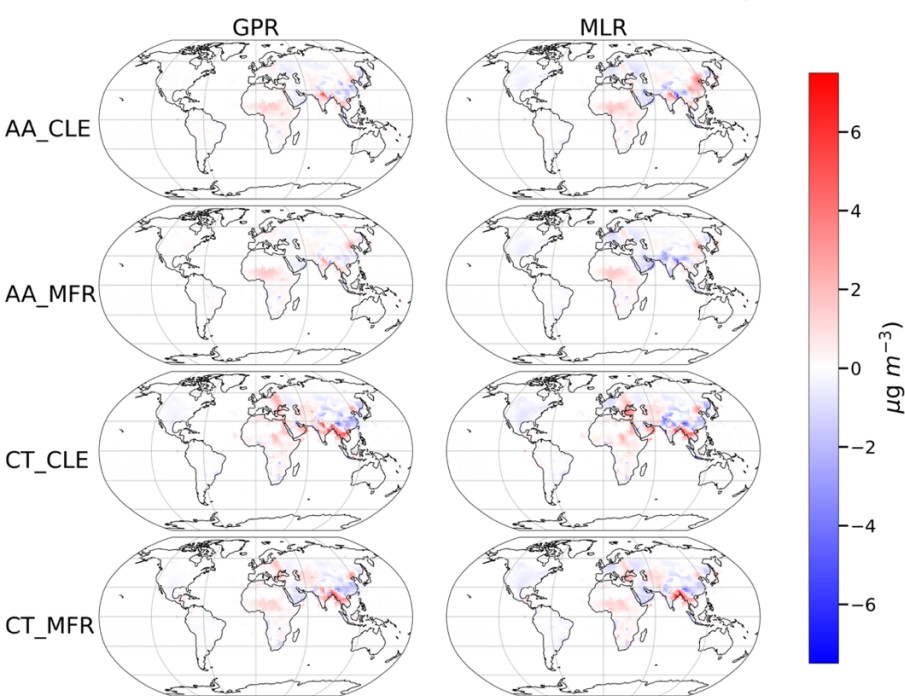

**Fig. 11 GPR and MLR emulator errors relative to GCHP-CAM simulated ΔPM$_{2.5}$ over the 4 IGSM-GAINS-TAPS scenarios at 2050 (relative to 2014).**

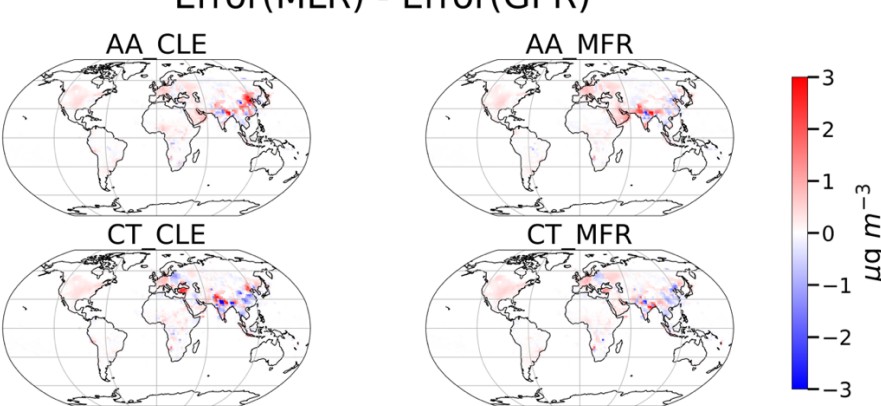

**Fig. 12 Difference in the absolute error (relative to GCHP-CAM output) between MLR and GPR emulation. Red (positive) indicate GPR is more accurate than MLR at the given grid cell, while blue (negative) indicates the opposite.**

For comparison, we train a multilinear regression (MLR) emulator with identical variables, geographic weighting and normalization schemes, and the performance metrics of the multilinear emulator is also shown in table 3. In all scenarios, the MLR estimator has a larger global MAE than GPR (by 0.05 μg m$^{-3}$ (19%) in CT_CLE to 0.10 μg m$^{-3}$ (40%) in AA_CLE).





However, in 3 out of 4 scenarios tested (expect AA_MFR), the GPR emulator has higher MB than the MLR emulator, though
the overall magnitudes of MB remain relatively small (within 0.1 μg m$^{-3}$).

Fig. 11 shows the spatial distribution of MLR and GPR error relative to their parent model (GCHP-CAM), and Fig. 12 shows
the difference in absolute values of such errors between MLR and GPR. In the 4 IGSM-GAINS-TAPS scenarios, GPR
predictions have less absolute error relative to the parent model than MLR in 57.6 % (CT_CLE) to 66.4% (AA_MFR) of the

grid cells. In all 4 scenarios, GPR outperforms MLR over the US (MAE = 0.05 – 0.14 μg m$^{-3}$ for GPR vs 0.17 – 0.29 μg m$^{-3}$
for MLR), western and southern Europe (MAE = 0.05 – 0.06 μg m$^{-3}$ for GPR vs 0.20 – 0.26 μg m$^{-3}$ for MLR), Middle East
(MAE = 0.16 – 0.56 μg m$^{-3}$ for GPR vs 0.40 – 0.72 μg m$^{-3}$ for MLR), South America (MAE = 0.08 – 0.14 μg m$^{-3}$ for GPR vs
0.14 – 0.16 μg m$^{-3}$ for MLR), and South Asia.

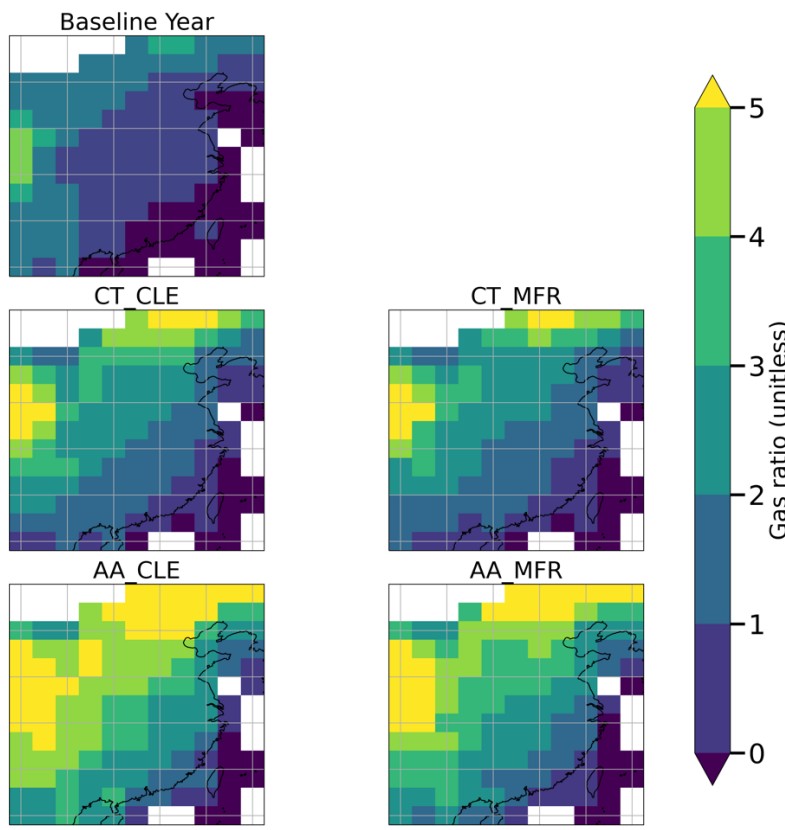

**Fig. 13. Gas ratio (GR) over China, which indicate secondary inorganic PM$_{2.5}$ sensitivity to NH$_3$ emissions. Secondary inorganic
PM$_{2.5}$ is weakly sensitive to NH$_3$ when GR < 0. 0 < GR < 1 indicate stronger sensitivity of secondary inorganic PM$_{2.5}$ to NH$_3$ emissions.
When GR > 1, sensitivity of secondary inorganic PM$_{2.5}$ to NH$_3$ emissions decreases as GR increases.**

In northern China, GPR is slightly less accurate than MLR on average under the MFR scenarios (by 0.31 μg m$^{-3}$ under
AA_MFR and 0.38 μg m$^{-3}$ under CT_MFR, measured by regional MAE), but considerably more accurate under CLE scenarios

(by 2.01 μg m$^{-3}$ under AA_CLE and 0.81 μg m$^{-3}$ under CT_CLE. We analyse the shifts in the chemical regime of secondary





inorganic aerosol formation calculating the Gas Ratio (*GR*) (Paulot and Jacob, 2014) over China at baseline year and under all 4 scenarios (Fig. 13):

$$GR = \frac{[NH_3]+[NH_4^+]-2[SO_4^{2-}]}{[HNO_3]+[NO_3^+]} \quad (7)$$

*GR* < 0 indicates that secondary inorganic PM$_{2.5}$ is weakly sensitive to NH$_3$ emissions through adding NH$_4^+$ to existing SO$_4^{2-}$

and HSO$_4^-$ ions. 0 < *GR* < 1 indicates that there is enough NH$_3$ to react with SO$_4^{2-}$, such that NH$_3$ and HNO$_3$ start partitioning into NH$_4$NO$_3$ aerosol, leading to strong sensitivity of secondary inorganic PM$_{2.5}$ to NH$_3$ emissions. In this regime, secondary inorganic PM$_{2.5}$ is more sensitive to NH$_3$ emissions. *GR* > 1 indicates that there is more than enough NH$_3$ to react with both SO$_4^{2-}$ and HNO$_3$, and PM$_{2.5}$ sensitivity to NH$_3$ emissions will weaken continuously as *GR* keep increasing beyond 1 (Ansari and Pandis, 1998). At the baseline year, *GR* over Norther China is largely between 0 – 1. Under all four scenarios, *GR* increases

beyond 1 over northern China. However, the increases in GR are the strongest under AA_CLE, followed by AA_MFR, while CT_CLE and CT_MFR have lower *GR* than the two AA scenarios. This indicates stronger shifts in secondary inorganic PM$_{2.5}$ sensitivity to precursor emissions relative to the baseline year (and therefore more non-linearity) under the two AA scenarios (especially AA_CLE) than the two CT scenarios, which is more well-captured by GPR than MLR.

The results in this sub-section show that GPR generally outperforms MLR. When emission changes could potentially trigger

non-linear aerosol chemistry, non-linear emulators can be significantly more accurate than linear emulators. This justifies the use of non-linear regression techniques (e.g. GPR) in developing air quality emulators.

### 3.3.2 Sensitivity to dispersion length scales

**Fig. 14 Changes in absolute error (relative to GCHP-CAM output) when no dispersion kernel is implemented. Red (positive)**
**indicates that turning off dispersion worsens the performance (increasing error), blue (negative) indicates the opposite.**

In addition to regression techniques, we also conduct 3 sensitivity tests of altering the dispersion length scales: 1) no dispersion; 2) halving the dispersion length scale, and 3) doubling the dispersion length scale. Fig. 14 shows the changes in absolute error of emulator prediction when the dispersion kernel is disabled, i.e. no geographic weighting is done. Turning off the geographic weighting scheme worsens the performance of the emulator, increasing the global MAE by between 0.31 (AA_MFR) and 0.88



(CT_CLE) µg m$^{-3}$, and locally absolute error by up to 29.1 µg m$^{-3}$. This decline in model performance is much larger than that

by switching from GPR to MLR, indicating the necessity of the geographic weighting scheme in our emulator.

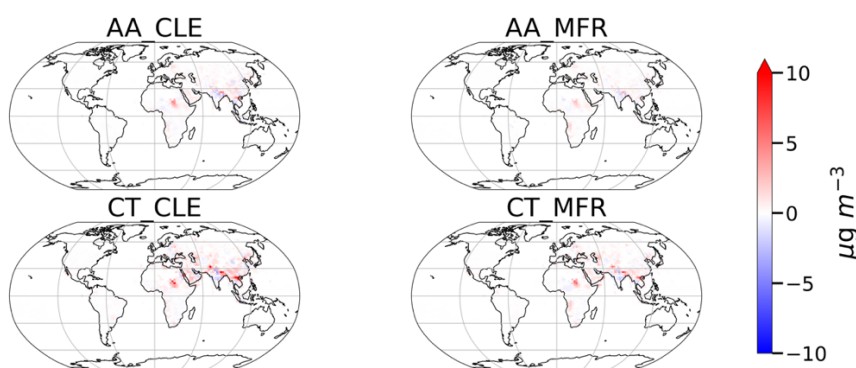

**Fig. 15 Changes in absolute error (relative to GCHP-CAM output) when the dispersion length scale is halved. Red (positive) indicates that turning off dispersion worsens the performance (increasing error), blue (negative) indicates the opposite.**

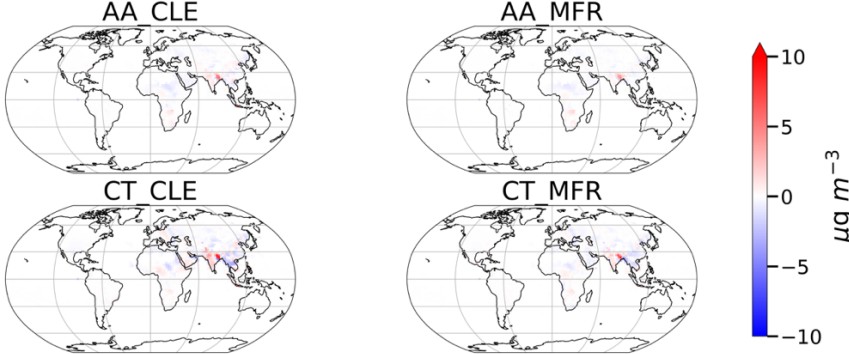


**Fig. 16 Changes in absolute error (relative to GCHP-CAM output) when the dispersion length scale is doubled. Red (positive) indicates that turning off dispersion worsens the performance (increasing error), blue (negative) indicates the opposite.**

Fig. 15 shows the changes in absolute error of emulator predictions when the dispersion length scale is halved. Halving the

dispersion length scale increases global MAE at all the scenarios by 0.05 (AA_MFR) to 0.21 (CT_CLE) µg m$^{-3}$, especially

over the Sahel and northern India, where absolute error increases by up to 13.2 µg m$^{-3}$. Fig. 16 shows changes in absolute error

of emulator predictions when the dispersion length scale is doubled. Doubling the dispersion length scale leads to minor

changes in global MAE across the 4 scenarios (-0.015 µg m$^{-3}$ under CT_CLE to +0.015 µg m$^{-3}$ under CT_MFR). The

geographic pattern of emulator performance changes is similar across different scenarios. After doubling the dispersion length

scale, the emulator performs better over the Sahel, China and Indochina by up to 4.8 µg m$^{-3}$ locally, but worse over India,

where the regional MAE increases by 0.28 (AA_MFR) to 1.01 µg m$^{-3}$ (CT_CLE), and local error increases by up to 4.05 to

9.67 µg m$^{-3}$ locally over the 4 scenarios.





The results of these sensitivity tests illustrate that the accuracy of our emulator is sensitive to the choice of dispersion kernel. While our choice of a set of globally uniform dispersion length scales provides a reasonable first-order approximation to emulate pollutant dispersion, performance of the emulator could conceivably be improved by regional, or even grid cell specific dispersion kernels. However, this will greatly increase the computing power required to train the model, and potentially require many additional global change scenarios to train and benchmark the model.

## 4 Comparison with AerChemMIP ensemble

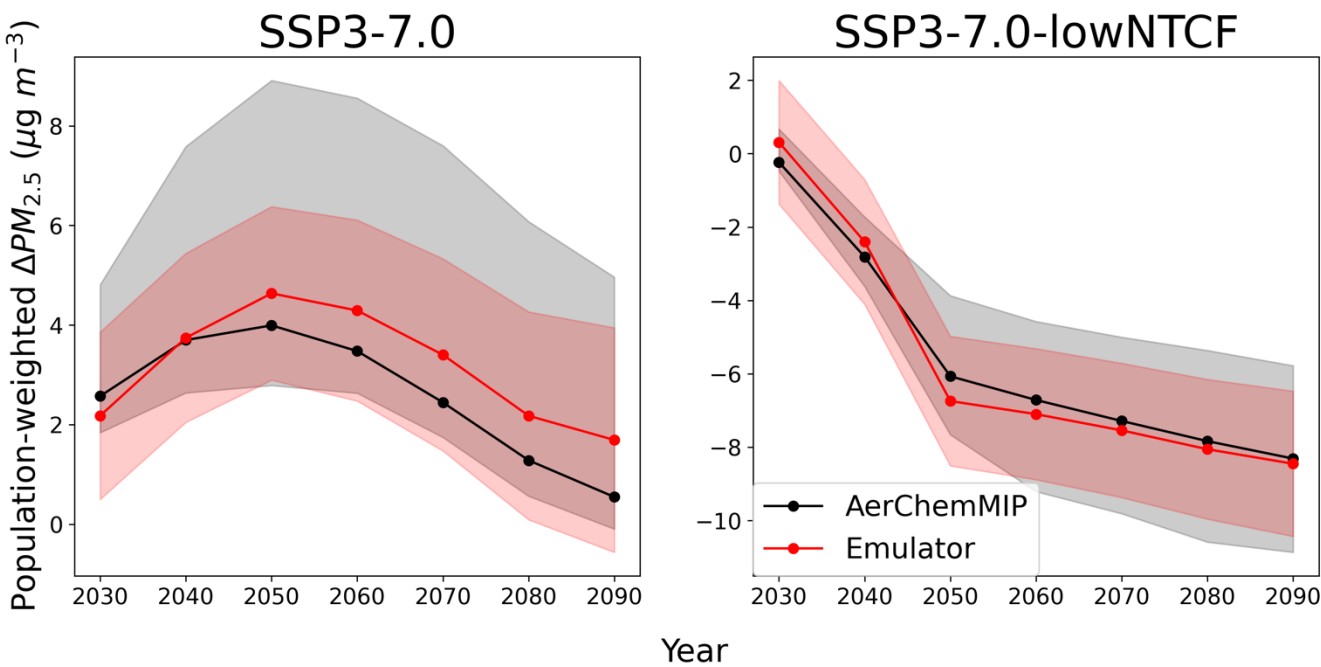

**Fig. 17 Changes in global decadal mean population-weighted anthropogenic PM$_{2.5}$ relative to 2015 – 2024 mean, predicted by the models in AerChemMIP ensemble and emulator with the shades indicate the range of uncertainty (ensemble range for AerChemMIP, 1 standard deviation for emulator), under standard SSP3-7.0 and the SSP3-7.0 low Near-Term Climate Forcer (SSP3-7.0-lowNTCF) scenario**

Fig. 17 shows the changes in of global population-weighted average anthropogenic PM$_{2.5}$ simulated by the models in the AerChemMIP ensemble and GW-GPR emulator over 2030 – 2090 under SSP3-7.0 and SSP3-7.0-lowNTCF scenarios, relative to 2015 – 2024 average. The solid lines represent the mean predictions, and the shaded areas represent the ranges of uncertainty (min/max for AerChemMIP ensemble, 1 standard deviation for emulator). Global population-weighted average ΔPM$_{2.5}$ from the emulator is within the range of AerChemMIP ensemble for both SSP3-7.0 and SSP3-7.0-lowNTCF scenarios over 2030 – 2090. The emulator predicted decadal mean global population-weighted average ΔPM$_{2.5}$ falls well within the range and differs by less than 1.15 µg m$^{-3}$ with the mean of AerChemMIP ensemble for all decades under both scenarios.





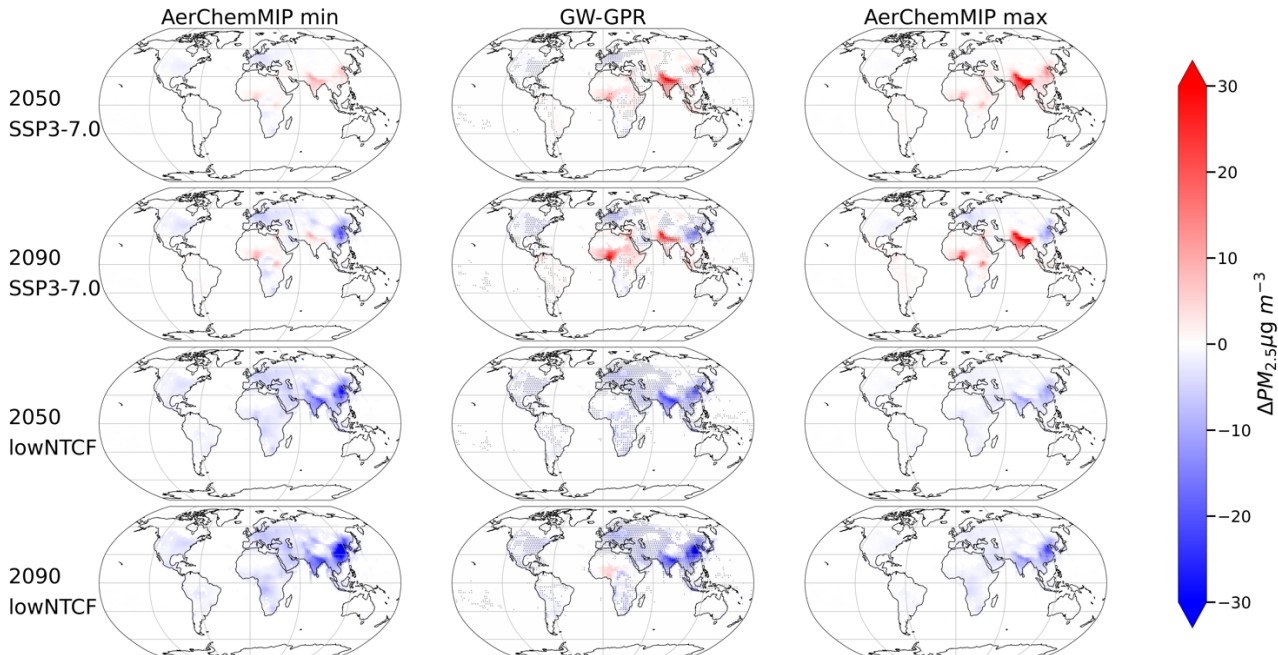


**Fig. 18 Multimodel minimum and maximum magnitude of ΔPM$_{2.5}$ simulated by models in AerChemMIP, and GW-GPR emulator predicted ΔPM$_{2.5}$ in 2050 and 2090 under SSP3-7.0 and SSP3-7.0-lowNTCF (abbreviated as lowNTCF in plot labels). Dots in the middle (GW-GPR) column indicate the grid cells where prediction of GW-GPR do not fall between the minimum and maximum of AerChemMIP simulation output.**

Fig. 18 show the multimodel minimum and maximum ΔPM$_{2.5}$ simulated by models in AerChemMIP and GW-GPR emulator predicted ΔPM$_{2.5}$ in 2050 and 2090 under SSP3-7.0 and SSP3-7.0-lowNTCF scenarios. Output from models in AerChemMIP is conservatively regridded to the same horizontal resolution for comparison. GW-GPR produces similar spatial patterns of ΔPM$_{2.5}$ as AerChemMIP models (e.g. large increases and decreases in PM$_{2.5}$ over northern China and northern India) in all four scenario-year combinations shown. Over major population centres in the northern Hemisphere (eastern North America,

Europe, northern India), GW-GPR emulator predictions of ΔPM$_{2.5}$ largely fall within the range of AerChemMIP model output. This contributes to the agreement of global population-weighted average ΔPM$_{2.5}$ between GW-GPR emulator and models in AerChemMIP (Fig.17).

However, there is systematic disagreement between GW-GPR and AerChemMIP model output over western Africa. Under SSP3-7.0, GW-GPR predicts 13.8 μg m$^{-3}$, while models in AerChemMIP predicts a 4.4 – 10.7 μg m$^{-3}$ increase in anthropogenic

PM$_{2.5}$ in 2090 over Nigeria. Under SSP3-7.0-lowNTCF, GW-GPR predicts a 3.0 μg m$^{-3}$ increase, while models in AerChemMIP predict a 1.8 – 3.0 μg m$^{-3}$ decrease in anthropogenic PM$_{2.5}$ in 2090 over Nigeria.

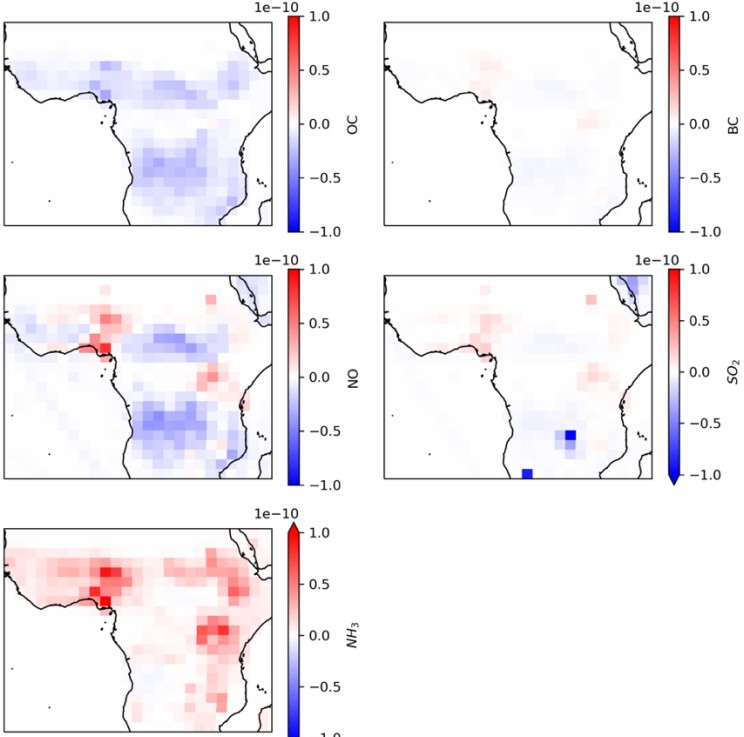

**Fig. 19 Changes in air pollutant emissions (kg m$^{-2}$ s$^{-1}$) in 2090 relative to 2020 over equatorial Africa under SSP3-7.0-lowNTCF**

To explore the potential sources of error and bias of GW-GPR emulator, we conduct more detailed analysis of $\Delta PM_{2.5}$ over

equatorial Africa in 2090 under SSP3-7.0-lowNTCF. Fig. 19 shows the changes in OC, BC, NO, SO$_2$ and NH$_3$ emissions over equatorial Africa. Over Nigeria, the magnitudes of NO (+145%), SO$_2$ (+180%) and NH$_3$ (+140%) emission changes are well beyond the range prescribed in our training set (±100%), which could lead to failure of machine learning algorithms. We also recognize GW-GPR has consistent positive biases over GCHP-CAM (Fig. 10) over equatorial Africa that cannot be effectively eliminated by switching to MLR (Fig. 11). This hints that the mismatch between regional pollutant transport patterns and

prescribed dispersion kernel could be another possible source of error of GW-GPR over equatorial Africa. Theoretically, the isotropic geographic weighting scheme only emulates of primary pollutants and precursors of secondary pollutants as a purely diffusive process, but not directly emulating the dispersion of secondary pollutants nor the advective components of pollutant dispersion. This might also contribute to the failure of the emulator to predict the changes in regional aerosol background, especially when the emission changes are highly spatially heterogenous.

The comparison between GW-GPR and AerChemMIP output further confirms ability of GW-GPR in predicting the spatiotemporal changes in anthropogenic PM$_{2.5}$ exposure under global change scenarios. However, GW-GPR predictions must be interpreted cautiously when the changes in emissions are well beyond that prescribed in the training set (±100%), or there is high level of spatial heterogeneity in pollutant emission changes within a region.





## 5 Discussion

In this work, we apply a classic emulator building workflow (carefully sampling the input space to create samples for training machine learning models) that has been widely applied in engineering (Alizadeh et al., 2020) to build a reduced-form global air quality model from a high-fidelity global 3-D chemical transport model, GCHP-CAM. Similar techniques (also often choosing Gaussian Process Regression as the machine learning algorithm) have been used for uncertainty analysis and parameter calibration in atmospheric chemistry modelling (Reyes-Villegas et al., 2023; Ryan and Wild, 2021; Wild et al.,

2020), and directly emulate air quality models at local and regional scales (Conibear et al., 2021; Vander Hoorn et al., 2022). Our work applies this approach for global change scenarios, where climate change, inter-regional chemical transport and discrepancies in chemical regimes pose another layer of challenges.

To address these challenges, there are a few unique features of the emulator architectures in comparison to other reduced-form global air quality models. We design the emulator to be usable for a wide range of integrated assessment modelling and policy

evaluation, where new scenarios or greenhouse gas concentration and pollutant emissions are routinely generated, but complementary atmospheric simulations are not always available. Therefore, rather than directly using the meteorological fields as input (e.g. Chen et al., 2023), we parameterize anthropogenic climate change intensity as a function of atmospheric $CO_2$ concentration, and use geographic weighting (Pisoni et al., 2018) to approximate the effect of chemical transport. Rather than exploring the source-receptor relationships between pre-defined regions, the emulator is trained at individual grid cell

level. Therefore, the accuracy of the emulator is not affected by different definitions of regions and sub-regional changes in spatial patterns pollutant emissions, which is important for application across different integrated assessment frameworks. This also allow us to tackle the non-linearity in the atmospheric chemical system by exploring more combinations of pollutant emission changes (more efficiently via Latin Hypercube Sampling) and machine learning (via Gaussian Process Regression).

By analysing emulator performance at grid cell level, we find the GW-GPR emulator successful in reproducing the global and

regional changes in $PM_{2.5}$ simulated by GCHP-CAM under 4 climate and air quality legislation (IGSM-GAINS-TAPS) scenarios and that from the AerChemMIP archive under SSP3-7.0 and SSP3-7.0-lowNTCF scenarios. We also find that the emulator may underperform when 1) the magnitude of pollutant emission changes is well beyond that prescribed in the training set (±100%); 2) the dispersion kernel ignores the advective component of pollutant transport, therefore misrepresents region-specific directional pollutant transport patterns; 3) the spatial pattern of pollutant emission changes is highly heterogenous

within a region. This points to some potential ways of improving the accuracy of the GW-GPR framework (e.g. fitting anisotropic dispersion kernels for each grid cells, expanding the training set), which can be explored in the future.

In addition to the mean, GPR also calculates the standard deviation of the prediction, which can be interpreted to characterize the statistical uncertainty of emulator output. As unforced climate variability directly contributes to the interannual variabilities of $PM_{2.5}$, previous studies recommend $10 - 20$ years of averaging to robustly detect the changes in $PM_{2.5}$ over contiguous

United States (Brown-Steiner et al., 2018; Garcia-Menendez et al., 2017; Pienkosz et al., 2019), especially when the signal to be detected is smaller or comparable in magnitude to the underlying unforced climate variability. Due to limitations in

computing time, we focus on exploring a wider range pollutant emission and climate change by generating each sample in the training set using one year of simulation, rather than running multiple years of simulation to generating robust signals amidst unforced climate variabilities for each set of perturbation experiment. This significant source of uncertainty, however, is captured by the uncertainty quantification algorithm of the GPR. Therefore, in addition to the magnitude, our emulator also provides the uncertainties in $\Delta PM_{2.5}$, which can be important in quantifying the overall uncertainties of health impacts of future air pollution (Saari et al., 2019).

In combination with the emission intensity projections from GAINS, TAPS can translate integrated assessment model output to spatially explicit air pollutant emission inventories. Combining with the GW-GPR emulator (Fig. 2), we can potentially produce gridcell-level projection of anthropogenic $PM_{2.5}$ changes for any climate and air quality integrated assessment scenarios within seconds, as demonstrated by our IGSM-GAINS-TAPS emulation exercise. This opens up the possibility for including air quality impacts within climate and sustainability decision making and scientific analysis. As climate projections move towards including scenario design as part of the uncertainty (Guivarch et al., 2022; Lamontagne et al., 2018; O'Neill et al., 2016, 2020), climate and global change scenarios generated will increase by orders of magnitude (Lamontagne et al., 2018; Shindell and Smith, 2019). Tools with proper balance between accuracy and computing resource requirements become more important in enabling uncertainty analysis and impact assessments. Our work shows the potential of machine learning techniques in enabling rapid and accurate global air quality assessment. Future work includes applying the $PM_{2.5}$ emulator to study more global change scenarios, improving and extending the emulator to calculate changes in other pollutants (e.g. $O_3$) and local climate forcing, and building software package and web interface to increase the accessibility of the emulator.

**Author contributions**

AYHW, NES and SDE conceptualized the study. EM developed the underlying CAM variant and produced the meteorological data. SDE developed and evaluated the GCHP-CAM model. AYHW conducted GCHP-CAM simulations, preparing air pollutant emissions for the IGSM-GAINS-TAPS scenarios, GW-GPR modelling, acquiring and processing AerChemMIP, Input4MIP data. AYHW analyzed the results and wrote the manuscript with input from NES and SDE. NES provided supervision and acquired funding for this project.

**Competing interests**

The authors have no competing interests



## Acknowledgement

This study is part of the Bring Computation to the Climate Challenge (BC3) project at MIT, which is supported by Schmidt
Sciences, LLC. and the MIT Climate Grand Challenges. The computation of this study was performed on the Svante computing
cluster, which is supported by MIT Center for Sustainability Science and Strategy. We acknowledge the climate modelling
groups participating in AerChemMIP and make the model output available through ESGF, and the funding agencies supporting
the above work. We thank Will Atkinson for developing TAPS, Emmie Le Roy for configuring GCHP-CAM, and other
members of the BC3 team and Selin group for their valuable insight.

## Code and data availability

The GW-GPR software is publicly available at https://github.com/ayhwong/GW-GPR, with a brief user guide in the repository.
A frozen version of the repository, which in addition contains the associated datasets required to reproduce the study and the
figures presented in this manuscript, is available on Zenodo (Wong, 2025) (https://doi.org/10.5281/zenodo.15484655). TAPS
is publicly available at https://github.com/watkin-mit/TAPS. The GAINS data used in this study were processed by Atkinson
et al. (2022). The raw GAINS scenario data are available at https://gains.iiasa.ac.at/models. The source code of GCHP 13.0.0
is available at https://github.com/geoschem/GCHP/releases/tag/13.0.0. Detailed descriptions and greenhouse gas emissions
for CT and AA scenarios can be found within the MIT Joint Program on the Science and Policy for Global Change 2023
Global Change Outlook (Paltsev et al., 2023). The unprocessed CAM and GCHP output are available by contacting the
corresponding authors.

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
