# Peer review of "A Geographically Weighted Gaussian Process Regression Emulator of the GCHP 13.0.0 Global Air Quality Model"

_EGUsphere, 2025_

## Author Comment (AC3)

\*Note: line numbers in the response reflect the line numbering in the changetracked PDF file

**Response to Referee 1**

We sincerely thank the reviewers for the constructive comments. Particularly, we appreciate and are eager to address the concerns about the clarity of presentation in our methodology section, and the motivation behind this study.

**Major comments:**

1. One major concern is the clarity on how the base modelling is described. Ultimately the output comes from GCHP but the meteorology for this comes from CAM which is provided with climate information from the MIT IGSM. There are emissions from GAINS via TAPs etc. It is unclear what the inputs and outputs are from each of these models and the configurations that are being used. This ultimately raises several questions revolving around the consistency between the emissions assumptions used for the driving climate / meteorology and the air pollution? This section of the methods needs to be significantly clarified.

Some of this is laid out in Figure 2 which should come earlier in the methodology section but there needs to be a significant effort to re-write the methodology is a more coherent and straight forwards way. I would start with the MIT IGSM, explain the configuration for that and the time-period of the model runs. Then what are the outputs from that and how are they used by the subsequent modelling tools. Then move onto the next model and explain its inputs and outputs.

Response: We thank the reviewer for the important comment. We agree that the inputs and outputs of each model components warrant clearer descriptions. We have rewritten the entire methodology section to address these issues raised by the reviewer. The new subheadings and their order reflect the reorganization based on the reviewer's suggestion, while the changes in individual sub-sections will be discussed in the more specific comments below:

- 2.1 MIT Integrated Global System Model (IGSM) and its coupling with Community Atmosphere Model (CAM)
- 2.2 GEOS-Chem High Performance model driven by CAM meteorological fields (GCHP-CAM)
- 2.3 Generating PM2.5 training data using GCHP-CAM
- 2.4 Training the Geographically Weighted Gaussian Process Regression (GW-GPR) Emulator
- 2.5 Cross validation of the GW-GPR emulator
- 2.6 Testing the emulator with IGSM-GAINS-TAPS combined air quality and climate legislation scenarios
- 2.7 Demonstrating the utility of the emulator using AerChemMIP data
- 2.8 Health impact calculation

New figures are added to better illustrate the MIT IGSM-CAM framework (new Fig. 1) and generation of GCHP-CAM training data (new Fig. 2). All the detailed changes in the content will be documented in **comment 18**.

**2. Reduced form modelling**

There is also a lack of clarity about how the gaussian process emulation is being undertaken. What is being predicted? Is it the PM2.5 or the individual tracers needed to calculate PM2.5? Is it all the GCHP tracers? In some places it suggests its PM2.5 but then there are discussions of the anthropogenic PM2.5 which would suggest that individual components have been predicted and then in other places its seem that the individual components are being predicted.

Response: We thank for reviewer for pointing out this ambiguity. We use **geographically-weighted pollutant emissions**, **global mean CH4 and CO2** concentration as predictors to predict **GCHP-CAM simulated changes in annual mean anthropogenic PM2.5 concentration relative to the 2014**. That is, we predict PM2.5 in the aggregate, not the individual components. We have included these passages in the revised methodology section:

L 178 – 184: Total PM2.5 mass (Eq. 1) is calculated... Anthropogenic PM2.5 mass (Eq. 2) is calculated...

```
Total PM_{2.5} = 1.1(NH_4^+ + NO_3^- + SO_4^{2-}) + BC + OC + fine mineral dust + SOA + 1.86(Sea Salt) (1)
Anthropogenic PM_{2.5} = 1.1(NH_4^+ + NO_3^- + SO_4^{2-}) + BC + OC (2)
```

In addition, we explicitly define the quantity we are modeling/emulating, and such definition of  $\Delta PM_{2.5}$  is consistent over the manuscript:

L 340-332:... The GCHP-CAM output we aim to emulate (changes in annual mean anthropogenic  $PM_{2.5}$  concentration relative to 2014 baseline ( $\Delta PM_{2.5}$ )) is calculated for each perturbation experiment.

What exactly are the input / output variables to make the prediction? A table of some sort would be useful here.

Response: We than the reviewer for pointing out this ambiguity. We now explicitly define the input and output of our emulator using an equation (Eq. 8):

L 436 – 440:... $d_{y,x}$  is the distance between grid cell y and x, and L is the dispersion length scale. Formally speaking, this implies at each grid cell x, our GW-GPR framework ( $f_x$ ) predicts GCHP-CAM simulated  $\Delta$ PM2.5 using  $\Delta$ Eweighted,x (the vector of  $\Delta$ Eweighted,x,i for all pollutants) atmospheric CH4 and CO2 concentration as input:

It would also be useful to know more explicitly what the training data consisted of. Data from what period to what period. Again, this isn't clear.

Response: We than the reviewer for pointing out this ambiguity. Our training data consists of output from 120 GCHP-CAM perturbation runs, following a Latin Hypercube configuration to effectively sample sensitivity of PM2.5 over a wide range of climate and air pollution emissions. We clarify the input for our 2014 baseline GCHP-CAM simulation as follows:

L 186 – 191: We conduct GCHP-CAM simulations for atmospheric composition for the year 2014 (with additional 3 months of simulations as spin up (output discarded) before the start of 2014) by applying IGSM-CAM simulated meteorology, anthropogenic emissions of air pollutants from the Community Emission Data System (Hoesly et al., 2018), and the monthly surface CH4 concentration derived by spatially kriging the observations from National Oceanic and Atmospheric Administration Global Monitoring Laboratory Cooperative Air Sampling Network for 2014. The resulting modelled total and anthropogenic PM2.5 concentrations serves as a baseline for subsequent comparisons.

We clarify the input and output for our GCHP-CAM perturbation runs, and how this produces the training data as follow:

Fig. 2 Schematic of generating the training set by GCHP-CAM perturbation experiments using an Iman-Conover (IC) transformed Latin Hypercube Sampled (LHS) scaling factors. The orange box represents existing modelling systems, purple boxes represent data sets, and blue boxes represents output of the perturbation experiments.

. . .

L 324 – 332: Fig. 2 summarizes the workflow of the perturbation experiments. A 1-year perturbation simulation, again with an extra 3 months before as spin-up, is performed for each pair of global scaling factors applied to 2014 anthropogenic air pollutant emissions and surface  $CH_4$  concentration. The  $CO_2$  concentration is directly applied to the calculate  $CO_2$  inhibition, and

the corresponding climate effects are represented through driving the simulation with the IGSM-CAM simulated meteorological data from the year with the closest  $CO_2$  concentration under "REF" scenario (e.g. A perturbation experiment having a  $CO_2$  concentration of 446 ppm is driven by the IGSM-CAM simulated meteorological data at 2030 under "REF" scenario, as 2030 has the closet  $CO_2$  concentration to 446 ppm among all years under "REF" scenario). The GCHP-CAM output we aim to emulate (changes in annual mean anthropogenic  $PM_{2.5}$  concentration relative to 2014 baseline ( $\Delta PM_{2.5}$ )) is calculated for each perturbation experiment as the training data set.

How is this a "geographically weighted" approach. This isn't clear in the description. What is meant by this?

Response: We than the reviewer for pointing out this ambiguity. We realize the term "geographic weighting" was introduced early in our manuscript when we discussion the Pisoni et al. (2017) emulator, without proper discussion of what that term means. We add this to our introduction:

L 109 – 110: Meanwhile, a geographically weighted (i.e. using a weighted sum of regional emission changes as predictors to represent pollutant transport process) linear regression emulator was shown to reproduce PM2.5 response to...

The term "geographically weighted" refers to the Gaussian dispersion kernel we use to calculate the effective emission changes for each air pollutant. We had used the term "dispersion kernel"/"blurring" interchangeably with "geographic weighting", which could be imprecise and underplay the essential role of the geographic weighting scheme in our emulator. Since dispersion kernel and blurring have their own more rigorous mathematical and academic definitions, "geographic weighting" is the most precise wording to descript our treatment to the pollutant emission fields. To avoid such confusion, we have now made our terminology more consistent, replacing the term "dispersion kernel" and "blurring" by "geographic weighting scheme" and "dispersion length scale/ $L_i$ " wherever possible:

L 432 – 435: To emulate the process of chemical transport of emitted species, an isotropic 2D Gaussian geographic weighting scheme is applied to calculate the effective air pollutant emission changes ( $\Delta E_{\text{weighted},x,i}$ ) at each grid cell x for each pollutant i...

Where  $\Delta E_{y,i}$  is the emission change of pollutant *i* all individual grid cells considered within the dispersion range  $y_k$ ...

L 441 – 445: The geographic weighting scheme is implemented by the Gaussian Blurring algorithm as in Scipy version 1.10.1 (Virtanen et al., 2020). The input variables are normalized by their corresponding global maximum value after the geographic weighting. Since the output variables are not geographically weighted, and  $\mu_2 = 0$  simplifies computation, the output variables are normalized by local mean and maximum at each grid cell. We note that some previous regional studies (e.g. Pisoni et al., 2017) have treated the parameters of the geographic weighting scheme as optimizable hyperparameters...

L 795 – 796: Fig. 15 shows the changes in absolute error of emulator prediction when the dispersion kernel is disabled, i.e. no geographic weighting scheme is disabled (i.e.  $L_i = 0$ )done.

L 883 – 885: The results of these sensitivity tests illustrate that the accuracy of our emulator is sensitive to the choice of  $L_i$  dispersion kernel. While our choice of a set of globally uniform dispersion length scales set of  $L_i$  provides a reasonable first-order approximation to emulate pollutant dispersion, performance of the emulator could conceivably be improved by regional, or even grid cell specific  $L_i$  dispersion kernels.

L 934 – 935: This hints that the mismatch between regional pollutant transport patterns and the prescribed geographic weighting scheme dispersion kernel could be another possible source of error...

L 977 – 981: ...2) the dispersion kernel geographic weighting scheme ignores the advective component of pollutant transport...(e.g. fitting anisotropic dispersion kernels geographic weighting schemes for each grid cell, expanding the training set)

**3. Improvement of the methodology**

There are many ways a reduced form model can be produced. This paper describes one way. However, there is only real utility in the methods if it is "better" than other approaches. Table 3 includes columns describing the results from a Multilinear regression emulator approach. It would be useful if this could be described in the methodology more and highlighted in more detail. How does Figure 6 look like with the MLR approach? Is the additional burden of the Gaussian Process system "worth it" compared to the simpler MLR approach?

Response: We thank the reviewer for this thought-provoking comment. We add the description of MLR model in Section 2.4:

L 451 – 456: Instead, we choose a globally uniform set of  $L_i$  as an approximation:  $L_{NOX}$ ,  $L_{NH3}$  and  $L_{NMVOC} = 1$  grid cell (cell = 2° latitude × 2.5° longitude);  $L_{SO2}$ ,  $L_{BC}$  and  $L_{OC} = 2$  grid cell;  $L_{CO} = 3$  grid cell. To understand the utility of non-linear regression techniques, we conduct an experiment by training Multiple Linear Regressors (MLR) (instead of GPR) to represent  $f_x$  using identical input. We find that GPR increases the accuracy of the emulator over MLR (particularly over regions where the changes in NOx and SO2 versus NH3 emissions are large enough to trigger non-linear responses in secondary inorganic aerosol formation) without incurring large increase computing resources required during the prediction process, therefore justifying the use of GPR over MLR. More details of the comparison are shown in Section 3.3.1

We think this comment invokes a two worthwhile dimension to unpack:

**1) Utility of our model**

We have highlighted how our approach is an improvement relative to the linearized and regionalized source-receptor approach (able to represent sub-regional changes and non-linear chemical processes). (L 107 - 111). However, we did not highlight how our emulator has different use case and strengths/weaknesses compared to reduced-order chemical transport models (e.g. InMAP). We now justify our new approach by discussing the strengths and

weaknesses of our approach relative to both major existing approaches of global reduced-form PM2.5 modeling (linear regionalized source-receptor matrices and reduced-order CTM):

L 75 – 77: ...resulting in a reduced-order chemical transport model that can be run faster and in higher resolution (Tessum et al., 2017; Thakrar et al., 2022), which are applicable for regional and global high-resolution (~1-4km) modelling with runtime of a few hundred CPU hours per model year (Tessum et al., 2017; Thakrar et al., 2022)....

L 1011 – 1014: ...climate and global change scenarios generated will increase by orders of magnitude (Lamontagne et al., 2018; Shindell and Smith, 2019), where the runtime required by reduced-order chemical transport models (a few hundred CPU hours per model year) could be a hurdle for large ensemble modelling, despite their advantage of having higher spatiotemporal resolution than global statistical emulators...

2) The choice of GPR (or any other machine learning algorithms) vs MLR

Some of the main advantages of our emulator (e.g. grid cell level input and output, account for climate change) can be achieved by MLR. Therefore, geographically weighted MLR could be another viable emulator, and can perform reasonably well when pollutant emission changes are small enough to not trigger significant changes in secondary inorganic aerosol formation.

We have comprehensively compared the accuracies between GRP and MLR in section 3.3.1. We find that GPR is more accurate when the changes in precursor emissions are large enough to shift the chemical regime of secondary inorganic aerosol formation. We conduct additional tests and find that the MLR emulator is only around 25% faster than GPR emulator. These indicate that GPR can provide accurate emulation at a wider range of pollutant emission changes than MLR without largely increasing the computing cost. We now include the computing speed factor in justifying the use of GPR emulation at the end of section 3.3.1:

L 765 – 769: The results in this sub-section show that GPR generally outperforms MLR. When emission changes could potentially trigger non-linear aerosol chemistry, non-linear emulators can be significantly more accurate than linear emulators without large increase in the computing power requirement. This justifies the use of non-linear regression techniques (e.g. GPR) in developing air quality emulators, especially given that GPR only requires 25% more runtime than MLR.

4. Overall, it is difficult to evaluate the work here as I can't really understand exactly what has been done. The paper is long and covers the development of the reduced form model and then some application. It would be worth thinking about whether these applications are useful. The health effects between the full model and the emulator are identical. This isn't a surprise given Figure 6. Similarly, it's not obvious to me that there is much utility in the work on the AerChemMIP comparison. The numbers calculated appear to be sensible, but do we learn much here? Given the performance of the emulator in Figure 17, this work appears to be just a statement about the performance of the GCHP/CAM/MIT IGCM system compared to the AerChemMIP models rather than the

emulator. There is a lot of work done here but the paper feels long. The main conclusions get lost in this, both by the range of topics discussed and the way that they are explained.

Response: We thank the reviewer for the constructive comment. The IGSM-GAINS-TAPS scenarios are both emulator evaluation and use case demonstration exercises. Since  $\Delta PM_{2.5}$  is calculated using GCHP-CAM, it can be used as a fair benchmark to gauge how faithfully can the emulator reproduce GCHP-CAM simulated  $\Delta PM_{2.5}$ . The accuracy of our emulator in these evaluation cases (as acknowledged by the reviewer) provides confidence for emulator users to apply our emulator as a "fast screening tool" to explore the possible  $PM_{2.5}$  air quality impacts from different climate scenarios, instead of having to run expensive chemistry-climate models that are especially infeasible when there is an ensemble of climate and air pollution control scenarios (e.g. Shindell and Smith, 2019).

On the other hand, the AerChemMIP exercise is more of a use case demonstration than a proper "evaluation", since the parent model of the emulator (GCHP-CAM) is not part of AerChemMIP. The comparison with other models exists to demonstrate that the emulator can be applied to estimate the time evolution of  $PM_{2.5}$  under different climate scenarios, which results in "sensible" (i.e. generally agrees with other mainstream chemistry-climate model) predictions in global changes in anthropogenic  $PM_{2.5}$ . We agree that the underlying differences between GCHP-CAM with AerChemMIP models (which we did not explicitly test) is probably the reason behind the agreement between the emulator with AerChemMIP models. But again, it only took less than a minute of CPU time to predict  $\Delta PM_{2.5}$  using the emulator, rather than hundreds of thousands (or even more) CPU hours using the standard chemistry-climate modeling framework.

We change the title of section 3 and 4 and some associated subheadings under these two sections to further highlight the difference between the IGSM-GAINS-TAPS and AerChemMIP exercise:

- 3 Comparisons with GCHP-CAM Evaluation of the emulator
- 3.2 Emulator cross validation and sensitivity
- 3.3 Comparing Emulator performance for over IGSM-GAINS-TAPS scenarios with GCHP-CAM
- 4 Evaluating utility: Comparison with AerChemMIP ensemble

We further clarify the conclusion of this exercise and emphasize the resource saving at the end of section 4:

L 940 – 943: The comparison between GW-GPR and AerChemMIP output shows that the GW-GPR emulator can generate predictions of  $\Delta PM_{2.5}$  that are within the range of output from mainstream chemistry-climate models at global scale, while requiring much less computational resources (at the order of 10-100 CPU seconds per scenario) to run. This confirms the utility ability of GW-GPR in predicting the spatiotemporal changes in anthropogenic  $PM_{2.5}$  exposure under global change scenarios ...

The question "but do we learn much here" is indeed extremely important to answer. If users were interested in a few custom-build climate/air quality scenarios, statistical emulation would not be that useful as the computational cost of running chemistry-climate models is manageable. But as we move towards using ensembles of climate and air quality scenarios to quantify uncertainties driven by human activities, running chemistry-climate (or even climate models) alone for each individual scenario would be infeasible, which in turn limits the utility of such an ensemble approach. Our emulator provides a much computationally cheaper way to translate pollutant emissions and GHG concentration from climate scenarios to  $\Delta PM_{2.5}$ , which is the most damaging outdoor air pollutant at global scale.

There are other more sophisticated architectures that can accurately emulate the higher-order statistics of PM2.5 and other major pollutants (e.g. O3), but they require meteorological fields as input (Li et al., 2025, 2022). Since simulating meteorological fields requires substantial computing power, such solutions do not satisfactorily reduce the computational power required to project ΔPM2.5. This guides our design philosophy: we want our emulator to be embeddable within integrated assessment modeling frameworks (coupling economic models with simple climate models), which means our emulator only require the output of integrated assessment models (pollutant emissions and GHG concentration) as input, while having small enough computing time requirement (<10 CPU seconds per scenario) that would not significantly slow down the integrate assessment workflow. This can be important for climate-air quality co-benefit research, ensemble modeling and user-friendly interactive tools for education and stakeholder engagement.

We agree that the motivation and design philosophy of our emulator is not advocated clearly enough, and therefore got a bit lost within the technical details of the paper. In our revision, we further emphasize the motivation behind our emulator design:

L 115 – 119: ... This results in a global reduced-form air quality anthropogenic PM2.5 model that can account for spatially heterogenous pollutant emission changes and non-linearity in atmospheric chemistry under multiple climate scenarios without requiring simulated meteorological fields as input, and provide robust uncertainty estimates, without drastically increasing the computational cost. These properties would make our reduced-form model a highly viable candidate for specific use cases (e.g. ensemble modeling, building interactive tools, embedding in integrated assessment workflows).

L 1011 – 1017: ...As climate projections move towards including scenario design as part of the uncertainty (Guivarch et al., 2022; Lamontagne et al., 2018; O'Neill et al., 2016, 2020), climate and global change scenarios generated will increase by orders of magnitude (Lamontagne et al., 2018; Shindell and Smith, 2019), where the runtime required by reduced-order chemical transport models (a few hours per model year) could be a hurdle for large ensemble modelling, despite their advantage of having higher spatiotemporal resolution than global statistical emulators; and statistical emulators that require meteorological fields as input would not be applicable as 3D climate simulations are too computationally expensive to be conducted for individual scenario. Tools with proper balance between accuracy, and computing resource and

input data requirements become more important in enabling uncertainty analysis, and impact assessments and human-Earth system feedback research...

5. The word "level" does a lot of work in this paper. It is used to mean "concentration" (pollution level), the vertical coordinate of the model grid (vertical level), the "degree" of global warming (level of global warming), a spatial scale (global level). It would be useful if there could be some specificity in the different words used here.

Response: We agree that the word "level" is used to refer to several distinct quantities. To remedy such ambiguity, we replace the word "level" with more specific wording or drop this term altogether wherever appropriate. E.g. for spatial scale, we use the word "scale", for pollution we use "pollutant concentration" or "pollutant emissions". Since there were so many ambiguous uses of "level" in the manuscript, we provide a few representative examples rather than an exhaustive list of changes we applied to the manuscript:

**For spatial scale:**

L 491 – 492: New GW-GPR models are built from the synthetic training set and predictions are made over the synthetic testing set at grid cell scale level.

L 582 – 583: Country-scale level baseline age- and cause-specific mortality rates are provided by the World Health Organization (WHO) (WHO, 2018).

L 690 – 691: ... $\Delta$ PM2.5 from GCHP-CAM, and the suitability of emulator output for public health impact calculation at global scale level.

For GHG and pollutant concentrations:

L 174 – 175: While BVOC and soil  $NO_x$  emissions are both calculated online (and therefore respond to climate and atmospheric  $CO_2$  concentration level)

L 193 – 194: ...GCHP-CAM perturbation experiments by scaling 9 input variables that affect  $PM_{2.5}$  and oxidant concentrations level

L 334 – 335: ...to relate the changes in pollutant emissions and climate with the corresponding changes in annual mean  $PM_{2.5}$  concentration level...

Redundant use of the word "level":

L 65: ... faster and easier to run while retaining a reasonable level of accuracy...

6. The words "high fidelity" is used in several places. It's not clear to me what this means. An alternative set of words should be used of more context given to what the authors mean. At a resolution of ~200km this is not a "high resolution" model.

Response: We thank the review for the comment. We agree that these words are unnecessary. We have deleted all the references to "high-fidelity" in our manuscript.

7. The paper title suggests an emulation of the whole model, but I think only the PM2.5 concentrations have been emulated. GCHP at this spatial resolution (~200km by 250km) isn't really an "Air quality" model, it's an "atmospheric composition" model or something like that but most people would think an "air quality" model would have a substantially higher spatial resolution.

Response: We thank the reviewer for the constructive suggestion. We agree that the title of our paper should be more specific. We have now changed the title of our paper to:

A Geographically Weighted Gaussian Process Regression (GW-GPR) Emulator of Anthropogenic PM2.5 the GCHP from the GEOS-Chem High Performance (GCHP) 13.0.0 Global Chemical Transport Air Quality Model

8. "Widely adopted." I don't think any of these techniques have been "widely" adopted. I would remove this comment.

Response: We agree with the assessment of the reviewer. We deleted these words from the abstract.

9. "Uncertainties resulting from both chemistry and climate variability" I understand what climate variability is. I'm not sure what chemistry variability. However, I'm not sure that the methodology used here addresses these issues. This should either be expanded to be clearer or removed.

Response: We agree that the uncertainty metric from the GPR is statistical, and therefore not specific to any particular sources of uncertainty. We change the associated sentence to:

L 28-30: To our knowledge, the GW-GPR emulator is the first global-scale emulator operating at grid cell level with explicit consideration of non-linearities in atmospheric chemistry, climate change, and provides predictive uncertainties resulting from both chemistry and climate variability.

10. Line 36. Sustainable development goals. I don't think air quality has been explicitly stated as part of the SDGs. There isn't an AQ SDG which is surprising. The AQ goals are given as sub, sub, SDGs (3.9.1 and 11.6.2).

Response: We agree that AQ is not explicitly an SDG, but rather a sub-goal within other SDG. We change the sentence to:

L 55 – 56: and addressing health and environmental impacts from ambient air pollution has been explicitly stated included as part of withe Sustainable Development Goals (goal 3.9.1 and 11.6.2) (United Nations, 2015).

11. Line 37. This suggests that the only way to evaluate the future air quality is through offline models. However, there are online ESM approaches which are in general the more used for this kind of long-term projections.

Response: We agree that our statement misses the ESM approach. We add a description of the ESM approach at the end of the sentence:

- L 59-61: ... as inputs to a chemical transport model to simulate the impacts on air pollutant concentration. Alternatively, the greenhouse gas (GHG) emission or concentration, and air pollutant emissions can be directly fed into chemistry-climate and Earth system models to further include the feedback between atmospheric composition and other components of Earth system.
  - 12. Line 54. "Frequently applied in recent science and policy studies" References should be given.

Response: We thank the reviewer for the constructive comment. The citations earlier in the sentence refer to the application of these techniques. We move the citation to make this clearer:

- L 77 79: These SR (Huang et al., 2023; Reis et al., 2022) and reduced-order (Camilleri et al., 2023) models have been frequently applied in recent science and policy studies (e.g. Huang et al., 2023; Reis et al., 2022, Camilleri et al., 2023)...
  - 13. Line 60. "Chemical regimes" What do the authors mean here? Ozone NOx-VOC limitations? Aerosol SO4-NO3-NH4 regimes?

Response: We thank for reviewer for the constructive comment. We refer to the secondary inorganic aerosol regime. We now make the clarification:

- L 95: ...when there are shifts in sulphate-nitrate-ammonium chemical regimes.
  - 14. Line 63. SOA is an important component of PM2.5

Response: We thank the reviewer for the constructive comment. We acknowledge the importance of SOA as part of  $PM_{2.5}$ . The sentences in the indicated line discuss the non-linearity with respect to precursor emissions. We emphasize that in this sentence that we are only focusing on inorganic  $PM_{2.5}$ :

L 97: ...different precursor emissions (NOx vs NH3 vs SO2 for inorganic PM2.5)...

We also add details relative to SOA in other part of our manuscript:

L 170 – 172: ...are assumed to be non-volatile. SOA formation follows a simple yield-based scheme that converts a fixed potion of isoprene, monoterpenes and other terpenoids into a lumped SOA precursor pool and another lumped SOA pool (Kim et al., 2015).

L 180 – 182: Anthropogenic PM2.5 mass is calculated by the above method, but only summing a subset of aerosol species (sulphate, nitrate, ammonium, BC and OC) while leaving out the omitting other aerosol species that are mostly from driven by natural non-industrial sources (dust, sea salt and SOA):

L 660-661: ...reflecting the fact that our emulator does not consider SOA in our definition of anthropogenic PM2.5.

15. Line 64. What are the direct vs indirect impacts of climate change on air pollution? Changes in the meteorology? Increased temperatures? Can this be more specific.

Response: We thank the reviewer for the constructive comment. The SR method essentially ignore all climate effects, whether direct or indirect. We also provide mechanistic examples of how climate affect PM:

L 98 – 100: Existing SR matrices and reduced-order models also often ignore the direct effects of climate change on air pollution (e.g. changing precipitation and associated wet deposition, temperature effects on gas-aerosol partitioning and oxidation chemistry) (Jacob and Winner, 2009).

16. Line 73. What is meany by "geographically weighted." This is used a lot in the paper but there isn't a definition of what this means.

Response: We thank the reviewer for the constructive comment. This is a good place to start defining what is meant by "geographic weighting". We add the following description:

L 109 – 110: Meanwhile, a geographically weighted (i.e. using weighted sum of regional emission changes as predictors to represent pollutant transport process) linear regression emulator

17. Why was Gaussian Process Regression chosen over other methods? What is it and why is an appropriate tool to use for this problem?

Response: We thank the reviewer for the constructive comment. We have included a comprehensive mathematical description of GPR in section 2.4. We agree that some high-level discussion about why GPR is chosen in the introduction will be useful. We add this discussion when referencing the regional GPR emulators:

L 334 – 336: We use Gaussian Process Regression (GPR) (Williams and Rasmussen, 1995) to relate the changes in pollutant emissions and climate with the corresponding changes in annual mean PM2.5 concentration at grid cell scale, because of its effectiveness in handling nonlinearity, good performance with small training set, and quantifying predictive uncertainties

As indicated earlier I found this difficult to understand. In the first paragraph the authors talk about GCHP but this study uses a chain of models to generate the PM2.5 concentrations under several climate and emissions scenarios. It is very hard to understand what they have done. This section should be re-written with an introduction to explain the system being used and then details of each model used given in turn. What information is being used by which models? How is the data transferred between these models. Figure 2 is a start for this. But the textual description should be clearer and more specific. What are the inputs into the MIT GCM? What are its configurations? What are the outputs? What are the inputs into CAM? What is the CAM configuration? And then then what are the outputs? What emissions is it using? etc

This whole section should be rewritten in a much more coherent way. Some more section headings to describe the MIT GCM, CAM, TAPS, GAINS, GCHP etc and the flow of information between them. Once the model framework has been outlined the experiments performed to develop the training data can be explained.

Response: We thank the reviewer for the constructive comment. The restructuring of the method section discussed in comment 1 reflects the suggestion of the reviewer: "discuss the modeling tools first, then talk about the experiments"

Apart from restructuring our method section, we also add descriptions and illustrations to better explain the data flow and input/output at each part of our modeling system:

For the IGSM-CAM framework that provides the GHG and pollutant emissions, GHG concentration, and the 3D meteorological fields for each climate scenario (L 137 – 153):

**2.1 MIT Integrated Global System Model (IGSM) and its coupling with Community Atmosphere Model (CAM)**

Fig. 1 Schematic of the IGSM-CAM modelling framework. Orange boxes represent modelling systems, purple boxes represent data sets. The red dashed box represents the MIT IGSM part of the framework

The climate scenarios used in this study are generated from the MIT IGSM framework (Fig. 1). The human system component of IGSM...As part of the scenario projection, EPPA provides regionalized and sectorized consumptions of different fuel types under the socioeconomic assumptions of each scenario. The yearly global average atmospheric GHG concentrations is then derived by driving the MIT Earth System Model (MESM) (Sokolov et al., 2018) with corresponding EPPA output. The associated greenhouse gas (GHG) and air pollutant emissions drive the MIT Earth System Model (MESM) (Sokolov et al., 2018) to simulate yearly global

average atmospheric GHG concentration, and concentrations of zonally averaged climate and near-term climate forcers (NTCF, e.g. aerosols, O3).

Since the output of IGSM is zonally-averaged, we simulate 3D meteorological fields using the IGSM-CAM framework (Monier et al., 2013) that links the IGSM to the National Center for Atmospheric Research Community Atmosphere Model (CAM) 3.1 (Collins et al., 2006). In this framework, CAM is driven by the IGSM output GHG concentrations, sea surface temperature anomalies, sea ice cover, and NTCF concentrations (Fig. 1). A pattern scaling algorithm is used to translate 2D NTCF output from IGSM to the 3D input fields required by CAM. The simulation outputs used in this study are described and evaluated in detail by Monier et al. (2015). IGSM-CAM is run with a horizontal resolution of  $2^{\circ} \times 2.5^{\circ}$  on 26 vertical layers up to 2.2 hPa.

For GCHP-CAM, which simulates atmospheric composition using GHG concentration, pollutant emissions and IGSM-CAM meteorological fields (L154 – 184):

2.2 GEOS-Chem High Performance model (GCHP) driven by CAM meteorological fields (GCHP-CAM)

We use the GCHP-CAM modelling system, which was described and evaluated in Eastham et al. (2023), to simulate global PM2.5 distribution, and its response to climate and pollutant emission changes. The modelling system is based on a customized version of GCHP 13.0.0 (The International GEOS-Chem User Community, 2024) that can be driven by the modelled meteorological fields of the Community Atmosphere Model (CAM) version 3.1 derived from the IGSM-CAM framework. Here we provide a brief description of the modelling system and specific setups for our work.

GCHP (Eastham et al., 2018) simulates  $PM_{2.5}$  by resolving the chemistry, transport, emission and deposition of relevant chemical species. Oxidant chemistry is simulated using a coupled VOC-CO-NOx-O3-aerosol-halogen chemical mechanism (Sherwen et al., 2016). GCHP is run at C48 (~200km) horizontal resolution with the same vertical layers with the IGSM-CAM simulations. The model output is remapped into a 2° latitude  $\times$  2.5° longitude horizontal grid conservatively (Jones, 1999).  $PM_{2.5}$  includes... non-volatile. SOA formations follow a simple yield-based scheme that convert a fixed potion of isoprene, monoterpenes and other terpenoids into a lumped SOA precursor pool and another lumped SOA pool (Kim et al., 2015).

Biogenic volatile organic compounds (BVOC) emissions follow Guenther et al. (2012) with isoprene inhibition by CO2 (Possell and Hewitt, 2011; Tai et al., 2013) included. Soil NOx emissions follows Hudman et al. (2012). While BVOC and soil NOx emissions are both calculated online (and therefore respond to climate and atmospheric CO2 concentration), mineral dust (Meng et al., 2021) and lightning NOx (Murray et al., 2012) emissions are held at 2014 level.

The model is driven with climate projections simulated by the IGSM-CAM (Monier et al., 2013) a modelling framework that links the MIT Integrated Global System Model (IGSM, Monier et al., 2018) to the National Center for Atmospheric Research (NCAR) Community Atmosphere Model (CAM) 3.1 (Collins et al., 2006). The simulations are described in detail in Monier et al. (2015). The global climate model is run from 2000 – 2100 with a horizontal resolution of 2° × 2.5° on 26 vertical layers up to 2.2 hPa. We choose the high-warming "REF" scenario (10 W/m²)

in 2100, resulting in 4.3 °C warming in 2080 – 2100 versus 1990 – 2009) to provide samples across a wide range of warming and CO2 concentration. The meteorological data from this elimate projection is processed into the format of the Modern Era Retrospective for Research and Analysis version 2 (MERRA-2) (Gelaro et al., 2017) meteorological fields used by native GCHP. GCHP is run at C48 (~200km) horizontal resolution with the same vertical layers with the CAM simulations. The model output is remapped into a 2° latitude × 2.5° longitude horizontal grid conservatively (Jones, 1999).

Anthropogenic emissions of non-greenhouse gas (GHG) air pollutants are from the Community Emission Data System (Hoesly et al., 2018). Biogenic volatile organic compounds (BVOC) emissions follow Guenther et al. (2012) with isoprene inhibition by CO2 (Possell and Hewitt, 2011; Tai et al., 2013) included. Soil NO4 emissions follows Hudman et al. (2012). While BVOC and soil NO4 emissions are both calculated online (and therefore respond to climate and atmospheric CO2 level), mineral dust (Meng et al., 2021) and lightning NO4 (Murray et al., 2012) emissions are held at 2014 level. The monthly surface CH4 concentration is prescribed by spatially kriging the observations from National Oceanic and Atmospheric Administration Global Monitoring Laboratory Cooperative Air Sampling Network at 2014 level. Scaling of anthropogenic emissions and atmospheric CH4 concentration in the training sets are described in the next section.

Aerosol concentrations...(fine mineral dust, sea salt). Total PM2.5 mass (Eq. 1) is calculated from the aerosol mass concentration...Anthropogenic PM2.5 mass (Eq. 2) ...

```
Total PM_{2.5} = 1.1(NH_4^+ + NO_3^- + SO_4^{2-}) + BC + OC + fine mineral dust + SOA + 1.86(Sea Salt) (1)
Anthropogenic PM_{2.5} = 1.1(NH_4^+ + NO_3^- + SO_4^{2-}) + BC + OC (2)
```

For the details about how to configure and run the GCHP-CAM experiments to generate the training set:

L 185 – 193:

2.3 GCHP CAM experiments Generating PM2.5 training data using GCHP-CAM We conduct GCHP-CAM simulations for atmospheric composition at 2014 (with additional 3 months of simulations as spin up (output discarded) before the start of 2014) by applying IGSM-CAM simulated meteorology, anthropogenic emissions of air pollutants from the Community Emission Data System (Hoesly et al., 2018), and the monthly surface CH4 concentration derived by spatially kriging the observations from National Oceanic and Atmospheric Administration Global Monitoring Laboratory Cooperative Air Sampling Network at 2014. The resulting modelled total and anthropogenic PM2.5 concentration serves as a baseline for subsequent comparisons.

To effectively sample the sensitivity...and global warming...

L 303 - 332:

2.3.2. Generating training dataset through GCHP-CAM perturbation experiments

| Variables                                            | Range             |
|------------------------------------------------------|-------------------|
| Air pollutant emission scaling factor                | 0 - 2             |
| Surface CH 4 concentration scaling factor | 0.5 - 2.5         |
| Atmospheric CO 2 level concentration      | 369.9 – 813.5 ppm |
| Meteorological year corresponding to                 | 2000 - 2100       |
| $CO_2$                                               |                   |

Table 1. Range of scaling factors and CO2 concentration, and the meteorological year corresponding to the CO2 concentration (under "REF" scenario) of the perturbation experiments. The range of atmospheric CO2 concentration is derived from the range of CO2 concentration between 2000 – 2100 under the "REF" scenario.

Fig. 2 Schematic of generating the training set by GCHP-CAM perturbation experiments using an Iman-Conover (IC) transformed Latin Hypercube Sampled (LHS) scaling factors. Orange box represent existing modelling systems, purple boxes represent data sets, and blue boxes represents output of the perturbation experiments.

120 sets of scaling factors for the 9 input variables (range displaced in table 1) are generated following a Latin Hypercube Sampling (LHS) (McKay et al., 1979) strategy...

We first run GCHP-CAM with meteorological fields from 1st Oct 2013 to 31st Dec 2014 and 2014 anthropogenic emissions, CH4 and CO2 concentration under the "REF" scenario to generate the baseline for comparison, with the first 3 months of model run discarded as spin-up (output not used). The rearranged LHS scaling factors sets are then linearly mapped to the range of inputs (Table 1), and each set corresponds to a 1-year perturbation simulation, again with an extra 3 months before as spin-up. Each of the perturbation simulations are driven by the globally scaled 2014 anthropogenic air pollutant emissions and surface CH4 levels. The scaling factor for CH4 (0.5 to 2.5) is chosen to enclose the range of CH4 concentration in 2100 projected by Meinshausen et al. (2020) over all scenarios.

Fig. 2 summarizes the workflow of the perturbation experiments. A 1-year perturbation simulation, again with an extra 3 months before as spin-up, is performed for each pair of global scaling factors applied to 2014 anthropogenic air pollutant emissions and surface CH4 concentration. The CO2 concentration is directly applied to the CO2 inhibition algorithm, and the corresponding climate effects are represented through driving the simulation with the IGSM-CAM simulated meteorological data from the year with the closest CO2 concentration under "REF" scenario (e.g. A perturbation experiment having a CO2 concentration of 446 ppm is driven by the IGSM-CAM simulated meteorological data at 2030 under "REF" scenario, as 2030 has the closet CO2 concentration to 446 ppm among all years under "REF" scenario). The GCHP-CAM output we aim to emulate (changes in annual mean anthropogenic PM2.5

concentration relative to 2014 baseline ( $\Delta PM_{2.5}$ )) is calculated for each perturbation experiments as the training data set.

For how we use the training set to build the GW-GPR emulator (L 334 - 483):

2.4 Training the Geographically Weighted Gaussian Process Regression (GW-GPR) Emulator

...an isotropic 2D Gaussian dispersion kernel geographic weighting scheme is applied to calculate the effective air pollutant emission changes ( $\Delta E_{\text{weighted},x,i}$ ) at each grid cell x for each pollutant i:

$$\Delta E_{\text{weighted,x,i}} = \sum_{y_k} e^{-d_{y,x}^2/2L_i^2} \Delta E_{y,i} (7)$$

Where  $\Delta E_{y,i}$  is the emission change of pollutant i within all individual grid cells considered within the dispersion range  $y_k$ ...Formally speaking, this implies at each grid cell x, our GW-GPR framework  $(f_x)$  predicts GCHP-CAM simulated  $\Delta PM_{2.5}$  using  $\Delta E_{\text{weighted},x}$  (the vector of  $\Delta E_{\text{weighted},x,i}$  for all pollutants) atmospheric CH4 and CO2 concentration as input:  $\Delta PM_{2.5}(x) = f_x(\Delta E_{\text{weighted},x}, CH_4, CO_2)$  (8)

For the IGSM-GAINS-TAPS modeling exercise to evaluate our emulator (L515 - 549):

2.2.1 IGSM-GAINS-TAPS combined air quality and climate legislation scenarios 2.6 Testing the emulator with IGSM-GAINS-TAPS combined air quality and climate legislation scenarios

...we evaluate the ability of the emulator in reproducing  $\frac{\text{GCHP GCHP-CAM}}{\text{CHP-CAM}}$  output anthropogenic  $PM_{2.5}$  over 2 climate...

...Future air pollutant emissions for each scenarios can be derived through the Tool for Air Pollution Scenarios (TAPS) (Atkinson et al., 2022) by considering climate (fuel consumption) and air pollution (emission intensities) policies independently...

We perform 10 years of GCHP-CAM simulations for each of the four IGSM-GAINS-TAPS scenarios with their respective anthropogenic air pollutant emissions, and CH4 and CO2 concentrations in 2050. The meteorological years (2031 – 2041 for AA and 2040 – 2050 for CT) are chosen to match the level of warming of our meteorological fields ("REF" scenario). The simulations are driven by IGSM-CAM meteorological fields from the "REF" scenario, with meteorological years (2031 – 2041 for AA and 2040 – 2050 for CT) chosen to match the CO2 concentration and TRF (i.e. AA has similar CO2 concentration and TRF at 2050 with "REF" over 2031 – 2041, and CT has similar CO2 concentration and TRF at 2050 with "REF" over 2040 – 2050). The same GCHP-CAM input anthropogenic air pollutant emissions, CH4 and CO2 concentrations in 2050 are fed into the GW-GPR emulator to estimate ΔPM2.5 under each scenario, which is then compared to the multiannual mean ΔPM2.5 simulated by GCHP-CAM (section 3.3).

19. Line 215. The choice of L appears somewhat arbitrary. Presumably it has something to do with the lifetime of the compound and some mean wind-speed component. Given a ~250km gridbox and say a ~5ms-1 wind. The timescale for air to be blown out of the box

is  $\sim$ 12 hours. Thus L=1 seems appropriate for NO and potentially for some of the shorter-lived VOCs. L=3 (36 hours) seems very short for CO. The authors should provide more of a chemical interpretation of this in their text or identify this as a weakness of their approach.

Response: We thank the reviewer for raising this important issue, which warrants clarification. We agree that the choice  $L_i$  seems arbitrary. We conduct another sensitivity simulation that use another set of  $L_i$  based on the atmospheric lifetime of the individual species, and find no improvement in emulator accuracy, while the generally larger  $L_i$  leads to large increase in computing cost of the geographic weighting scheme. Therefore, we choose to retain our original choice of  $L_i$ , and explore the limitation from this approach. We add the description this sensitivity test to our manuscript:

L 449 – 481: Instead, we choose a globally uniform set of  $L_i$  as an approximation:  $L_{NOX}$ ,  $L_{NH3}$  and  $L_{NMVOC} = 1$  grid cell (cell = 2° latitude × 2.5° longitude);  $L_{SO2}$ ,  $L_{BC}$  and  $L_{OC} = 2$  grid cell;  $L_{CO} = 3$  grid cell...

In addition, we train the GW-GPR emulator by choosing another set of globally uniform  $L_i$  based on the typical atmospheric lifetime of individual pollutants, which results in larger  $L_i$  for most pollutants. However, we find that such set of  $L_i$  increases the computing cost of the Gaussian blurring without providing improvements in emulating  $\Delta PM_{2.5}$ . Therefore, we retain the choice of our original set of relatively small  $L_i$ , which provides enough distinction of dispersion length scales of different pollutants without invoking considerable additional computational cost...

And the result of this sensitivity test is shown in Section 3.3.2:

L 793 – 795: In addition to regression techniques, we also conduct  $\frac{3}{4}$  sensitivity tests of altering  $L_i$  the dispersion length scales: 1)  $L_i = 0$  (no geographic weighting) no dispersion; 2) halving  $L_i$  the dispersion length scale, 3) doubling  $L_i$  the dispersion length scale, 4) directly using atmospheric lifetime of pollutants to approximate  $L_i$ ...

L 838 – 882: In the final sensitivity experiment, we assume  $L_i$  (in km) to be approximately equal to  $10-20 \times$  atmospheric lifetime of pollutant i ( $\tau_i$ ) (Li and Cohen, 2021). Anthropogenic NOx and NH3 have very short  $\tau$  (within a few hours) (Dammers et al., 2019; Lange et al., 2022). Also, HNO3 (the main oxidized form of NOx) (Muller et al., 1993) and NH3 (Schrader and Brümmer, 2014) deposit rapidly. However, the secondary inorganic aerosol formed from anthropogenic NOx and NH3 have longer  $\tau$  (3 – 5 days) (Bian et al., 2017). Balancing these two factors, we assume  $L_{NOx}$  and  $L_{NH3}$  = 1 grid cell. SO2 has longer  $\tau$  (4 – 12 hours for point sources (Fioletov et al., 2015), and 1 – 1.5 days at regional and global scale (Chen et al., 2025; Hardacre et al., 2021; Lee et al., 2011)) and lower deposition velocity (Hardacre et al., 2021) than NOx and NH3. Therefore, we assume  $L_{SO2}$  = 3 grid cells. Anthropogenic NMVOC that has the most significant contribution to photochemistry and oxidation chemistry (e.g. xylene, toluene, ethylene, propylene) (Gu et al., 2021; Ran et al., 2011) typically have  $\tau$  of a few hours to 2 days (Franco et al., 2022; Tiwari et al., 2010; Trentmann et al., 2003). Therefore, we assume  $L_{VOC}$  = 2 grid cells. As recent studies suggests that  $\tau_{BC}$

Fig. 18 Changes in absolute error (relative to GCHP-CAM output  $\Delta PM_{2.5}$ ) when an alternate set of dispersion length scales ( $L_{NH3} = L_{NOx} = 1$  grid cell,  $L_{VOC} = 2$  grid cells,  $L_{SO2} = 3$  grid cells,  $L_{BC} = L_{OC} = 7$  grid cells,  $L_{CO} = 15$  grid cells, informed by the atmospheric lifetime of each pollutant i ( $\tau_i$ )) is applied to train the GW-GPR. Red (positive) indicates that using the alternate set of  $L_i$  worsens the performance (increasing error), blue (negative) indicates the opposite.

Fig. 18 shows the changes in absolute error of emulator predictions with the set of  $L_i$  described above. The global performance metrics (MAE =  $0.21 - 0.42 \,\mu g \, m^{-3}$ , MB =  $0.06 - 0.11 \,\mu g \, m^{-3}$ ) are very similar to the metrics obtained by training the GW-GPR emulator with the default set of  $L_i$ . The regional pattern of changes in emulator accuracy is consistent over the 4 scenarios tested. Using this alternate set of  $L_i$  reduces the error over southern China by up to 4  $\mu g \, m^{-3}$ , while increasing the error over western Africa and central Asia by up to 4  $\mu g \, m^{-3}$ . Over northern India and Bangladesh, the emulator error could locally increase or decrease by up to 5  $\mu g \, m^{-3}$ . However, the generally larger  $L_i$  increases the runtime of the geographic weighting scheme (around 15 seconds), while only 0.6 second is required to run the geographic weighting scheme using our default choice of  $L_i$ . Given the GW-GPR emulator can finish its prediction within 10 seconds for each scenario, such a large increase in runtime without consistent global improvement in emulator performance is not justified.

20. What is a random variable? Is this a normally distributed variable? This is a bit confusing as it could be construed as a variable containing random numbers, but I don't think this is what is meant. What is N in equation (1)? Are the PM2.5 surface concentrations and the input variables normally distributed? My guess is that they are not and many of them are likely log normally distributed. Does this matter?

Response: We thank for reviewer for the interesting question. In statistical emulation, the predicted variable (in our case  $\Delta PM_{2.5}$ ) is formulated as a distribution due to the associated predictive uncertainty. In this case, N refers to a multivariate normal distribution. GPR assumes the input and output variables are jointly normally distributed, so that the prediction (more precisely, predictive mean) (Eq. 4. in the revised manuscript) and uncertainty (predictive standard deviation, Eq. 5 in the revised manuscript) has a clean analytic expression, reducing the whole non-linear regression problem to finding an optimal set of parameters for the covariance

function (Eq. 6 in the revised manuscript) given the training data. i.e. The normal distribution assumption is a commonly acknowledged working hypothesis facilitating the computation, and the mathematics required to test this assumption is beyond the scope of this paper. We hope these changes in the manuscript would make the mathematics a bit clearer:

- L 352 429: We use a sum of anisotropic (in the input space, not the physical distance described below) functions GP kernels to represent the nature of our problem. These are (smooth functions (rational quadratic function) with unknown points of chemical regime change + local interactions among variables (Matern 3/2 function) + noise from climate variability (white noise). "Training" the GPR essentially means optimizing the parameters of the covariance function k against the training data set...
  - 21. What bits of information are being used here? Exactly what is being predicted and with what information?

Response: We thank the reviewer for the constructive comment. We make the following clarification:

L 432 – 440: ...an isotropic 2D Gaussian dispersion kernel geographic weighting scheme is applied to calculate the effective air pollutant emission changes ( $\Delta E_{\text{weighted},x,i}$ ) at each grid cell x for each pollutant i:

$$\Delta E_{\text{weighted,x,i}} = \sum_{y_k} e^{-d_{y,x}^2/2L_i^2} \Delta E_{y,i} (7)$$

Where  $\Delta E_{y,i}$  is the emission change of pollutant i within all individual grid cells considered within the dispersion range  $y_k$ ...Formally speaking, this implies at each grid cell x, our GW-GPR framework  $(f_x)$  predicts GCHP-CAM simulated  $\Delta PM_{2.5}$  using  $\Delta E_{\text{weighted},x}$  (the vector of  $\Delta E_{\text{weighted},x,i}$  for all pollutants), and atmospheric CH4 and CO2 concentration as input:  $\Delta PM_{2.5}(x) = f_x(\Delta E_{\text{weighted},x}, CH_4, CO_2)$  (8)

22. I might move this description (section 2.4) to be in the section 4 as it feels disjointed in the flow of the text.

Response: We thank the reviewer for this suggestion. We believe after the reorganization, the description of AerChemMIP exercise (now section 2.7) should no longer feel out of the place.

23. Section 3.2 It would be useful there could be some description at the start of how the evaluation is going to take place, what is going to be contained in this section. As described early the metrics are only useful if they are compared to an alternative method of reduced model generation. It seems like this has been done but it would be useful if this could be the basis for the evaluation? Is the new approach better than the old, rather than providing metrics of the performance of the new model in isolation.

Response: We thank the reviewer for this question. Our other "benchmark" would be replacing GRP by MLR. However, as discussed above (comment 3), there are other factors (better performance in under non-linearity, availability of predictive uncertainty) that led us to choose GPR over MLR other than the performance metrics in cross-validation. Presenting the crossvalidation result from MLR would be distracting and confusing for readers. Therefore we only present the cross-validation result from GPR here. We add description of the material at the start of the sub-section, so that the reader can refer to section 2.5 if they want to know how the result at this sub-section is generated and what it means:

L 621: In this sub section, we discuss the result of the cross validation and sensitivity test outline in section 2.5. Fig. 7 shows...

24. Line 293. This says that Figure 6 shows the comparison with the Delta Anthropogenic PM2.5 but the figure caption text says that it just Delta PM2.5.

Response: We thank the reviewer for this question. In this study, we define  $\Delta PM_{2.5}$  as changes in anthropogenic  $PM_{2.5}$ . We added this to the figure caption:

L 610 – 612: Fig 7. 2D histogram from the grid cell by grid cell comparison between changes in annual mean anthropogenic PM2.5 ( $\Delta$ PM2.5) predicted by the GW-GPR emulator ( $\Delta$ PM2.5, GW-GPR) and that simulated by GCHP-CAM ( $\Delta$ PM2.5, GCHP) from the 10-fold random subsampling cross-validation.

It's not clear over what time period this is run for? What is the calculated delta between?

Response: We thank the reviewer for this question. This is a 10-fold cross validation (where 80% of the training data is randomly selected to train a synthetic version of the model, while the remaining 20% of the training data is held out as synthetic testing data, and repeat this procedure for 10 times). We now refer the readers to section 2.5 for explanation (see last comment).

25. Line 300. Why does the standard deviation of the prediction and the MAE between the prediction and GCHP indicate that emulator SD is an appropriate measure of chemical and climate uncertainty. This should be explained in more detail.

Response: We thank the reviewer for this question. The emulator SD is from eq. 5, which is computed jointly with the emulator prediction mean (eq. 4) (see comment 20) and measures the predictive uncertainty of the emulator. It is naturally related the range/error of prediction (we state explicitly the text that "the emulator output standard deviation which can characterize the uncertainty of emulator prediction"). We can see that the standard deviation can get confused as being calculated from some sample populations. We now reference Eq. 4 to clarify that we are referring to the predictive uncertainty:

L 625 – 627: Fig. 7 8 shows the spatial distribution of grid cell level MAE of the GW-GPR emulator, and the emulator output standard deviation from Eq. 4 (which can characterize the predictive uncertainty of emulator output).

We agree that the "chemical and climate uncertainty" is confusing (see comment 9). Rather than "indicating", the similarity between MAE and predictive SD "confirms" that predictive SD is a good measure of emulator uncertainty. We now remove the reference to "chemical and climate uncertainty" in both the figure caption and main text:

L 618 – 620: Fig. 8 The mean absolute error (MAE) of emulator prediction against the parent model (GCHP-CAM), and the average standard deviation of emulator predictions from Eq. (4) (indicative of predictive uncertainties from climate variability and chemistry) at grid cell level (240 data points at each grid cell)

L 628 - 630: The emulator output standard deviation have similar magnitudes and spatial distributions (spatial  $R^2 = 0.99$ ) as MAE, indicating confirming that emulator output standard deviation is an appropriate measure of chemical and climate the uncertainties of emulator predictions relative to the parent model.

26. Figure 9. This is very small and hard to read. I can't see any dots in this figure, but they are described in the caption. The colour scale isn't very useful as to my eye as everywhere seems to show a reduction other than a possibly over Bangladesh.

Response: We thank the reviewer for the comment We have reduced the range the color scale from 50  $\mu g$  m-3 to 30  $\mu g$  m-3 and replaced the dots with hatches. We hope these changes will increase the readability of the Fig. 9 (Fig. 10 in the revised manuscript):

Fig. 10 Spatial patterns of GCHP-CAM and emulator predicted  $\Delta PM_{2.5}$  for each of the 4 IGSM-GAINS-TAPS scenarios at 2050 (relative to 2014). Only results in grid cells with population density > 1 person km-2 are shown. The <del>dots</del> hatches show where GCHP-CAM output does not fall within the 95% confidence interval of emulator prediction.

27. Table 3. The description of the multilinear approach is pages ahead in the document. It should be in the methods section and explained properly.

Response: We thank the reviewer for the comment. We now include description of the MLR in section 2.4 (see comment 3).

28. Is it clear why Section 3.3.2 is that rather than Section 3?

Response: We thank the reviewer for this question. Section 3.3 describes the GW-GPR performance of the IGSM-GAINS-TAPS experiments. The most important content is the performance of our baseline GW-GPR, while the MLR (3.3.1) and  $L_i$  (3.3.2) are "sensitivity experiments" the help exploring the strengths and limitations of baseline GW-GPR, and therefore written as 2 subsections after describing the performance of the baseline GW-GPR.

29. I think the workflow has been applied to a number of applications outside of Engineering. There are a number of examples of similar approaches in atmospheric composition research.

Response: We thank the reviewer for this comment. We agree with the reviewer that neither geographically weighted regression, Gaussian Process regression, nor extracting the sensitivity of pollutant concentration to precursor emissions through carefully designed sensitivity simulations are new approaches to model air pollution. This is evident at the start of our introduction, by how we acknowledge the previous usage of such workflow in atmospheric chemistry: "Similar techniques (also often choosing Gaussian Process Regression as the machine learning algorithm) have been used for uncertainty analysis and parameter calibration in atmospheric chemistry modelling (Reyes-Villegas et al., 2023; Ryan and Wild, 2021; Wild et al., 2020), and directly emulate air quality models at local and regional scales (Conibear et al., 2021; Vander Hoorn et al., 2022)." (L 955 – 958)

However, we are confident that we are among the first to creatively combining geographically weighting and machine learning regression techniques to tackle the challenges in emulating a global atmospheric chemistry model. This results in our emulator being able to provide improvement upon the linear source-receptor framework (e.g. grid cell-level input and output, able to handle non-linearity), while still costing much less computing resources than reduced-order chemical transport models.

In the conclusion, we further highlight that we are not merely applying the standard Gaussian Process surrogate techniques to atmospheric chemistry modeling. Rather, we extend this approach by combining the classic workflow with other features (e.g. geographic weighting scheme, using CO2 to parameterize global warming):

L 959 – 960: ...Our work applies extends this approach for global change scenarios, where climate change, inter-regional chemical transport and discrepancies in chemical regimes pose another layer of challenges.

**Response to Referee 2**

We sincerely thank the reviewers for the constructive comments. Here are our responses, and the corresponding revisions in our manuscript:

**Major comments:**

1. The authors make several assumptions in generating training data with GCHP-CAM including using CO2 levels instead of total radiative forcing to parameterize total radiative forcing in the 21st century. Uncertainty that arises from these assumptions is not addressed when presenting ΔPM2.5 and Δmortality results.

Response: This is an important point. We agree with the reviewer that the related uncertainties deserve more explanation and elaboration than what we presented in our manuscript. There are a few facts and assumptions behind the logic of using CO2 to parameterize climate impacts on anthropogenic PM2.5:

- 1. Response of global mean temperature to well-mixed greenhouse gases  $(\Delta \overline{T})$  is a function of changes in total radiative forcing from well-mixed greenhouse gases  $(\Delta F)$ .
- 2. The local responses of temperature and precipitation (which directly influence PM2.5 level through gas-aerosol partitioning and wet deposition) to well-mixed greenhouse gas forcing can be largely predicted by changes in global mean temperature, and these local relations ("patterns") are time-invariant (Lütjens et al., 2025).
- 3. Therefore, total radiative forcing can parameterize the impacts of climate change on PM2.5. The local relationships can be learned statistically, and the statistical uncertainties is quantified by standard deviation output from Gaussian Process Regression.
- 4. In a lot of climate scenarios, CO2 dominates both the overall magnitudes and trends in long-lived greenhouse gas radiative forcing in the 21st century (Meinshausen et al., 2020). Therefore, we can further parameterize total radiative forcing as an atmospheric CO2 level.

This simplifies our statistical model and increases its applicability, since CO2 level is among the most widely available climate forcing variables. A closer examination also reveals two potential weaknesses of our approach:

- 1. Our approach fails when climate change is not a sole function of well-mixed greenhouse gas forcing, therefore cannot capture the effects of local climate forcing (e.g. land use change, aerosol forcing), overshoot scenarios, etc. (Basically, this is equivalent to the cases where pattern scaling approaches typically fail in predicting future climate) (Giani et al., 2024)
- 2. Our approach fails when the trend of CO2 emission is decoupled with that of other greenhouse gases.

In addition, we trained another model adding greenhouse gas (CO2+CH4+N2O+CFC-11+CFC-12) radiative forcing as a predictor, on top of the 9 predictors presented in our manuscript. The

changes in error relative to GCHP-CAM after including ERF as an additional predictor for each IGSM-GAINS-TAPS scenarios is plotted as follows:

We find that including ERF as another predictor does not change the performance of the model significantly (global MAE increases by  $0.002-0.006~\mu g~m^{-3}$ ). Given that CO2 concentration is the most widely available global warming indicator, we maintain the use of CO2 level to parameterize global warming level. However, there are also other ways are also other ways (e.g. cumulative GHG emissions, combination with climate emulators) to parameterize climate effects that warrants further explorations beyond the scope of our paper.

Therefore, we add a whole section ("2.3.1 Parameterizing global warming") in our revision to provide a clearer explanation to our approach, and the associated uncertainties involved:

Representing the direct impacts of climate change is an important aspect of building climate-

L 244 - 276:

**2.3.1 Parameterizing global warming**

aware reduced-form atmospheric composition models. However, unlike the other 8 perturbed variables, global warming cannot be directly implemented as a scaling factor in GCHP-CAM. Some recent studies achieve this goal by including 3D meteorological fields from climate model output as predictors (e.g. Li et al., 2025, 2022). However, this could limit the utility of the model to scenarios where climate model outputs are archived in a correct format. We use a simpler parameterization of climate effects in our emulator to expand its applicability. We use the GHG concentration and IGSM-CAM simulated meteorological fields from its highwarming "REF" scenario (10 W/m2 in 2100, resulting in 4.3 °C warming in 2080 – 2100 versus 1990 – 2009) to provide samples across a wide range of global warming and GHG concentration from 2000 – 2100. While climate change can affect PM2.5 through pathways other than simply warming (e.g. precipitation, regional stagnation, mixing depth) (Jacob and Winner, 2009), changes in meteorological variables due to well-mixed GHG forcing can usually be parameterized as spatially-varying functions ("patterns", which are specific to individual climate models) of global mean temperature (e.g. Lütjens et al., 2025), which is a function of total radiative forcing by GHG (with climate sensitivity specific to each climate/Earth system model). Therefore, the effects of GHG-forced climate change on PM2.5 can be parameterized by total radiative forcing by GHG (TRF), which largely simplifies the statistical modeling and increases its applicability by not requiring meteorological variables as inputs. This implies the relation

between GHG-forced climate change and PM2.5 can be statistically learned by regression algorithms at each grid cell, when TRF is included as one of the input variables. In our emulator, we further parameterize TRF as atmospheric CO2 concentration. In many climate scenarios, CO2 is projected to dominate (68 – 85%) TRF and its trend in the 21st century (Meinshausen et al., 2020). In addition, atmospheric CO2 concentration also directly affects isoprene emission, which could affect atmospheric oxidant (e.g. OH, O3) (e.g. Tai et al., 2013), and therefore potentially secondary inorganic aerosol formation.

Parameterizing climate effects as TRF/CO2 concentration allows our statistical model to include climate effects without explicitly requiring meteorological fields as input, which makes our emulator easy to integrate within the workflow of integrated assessments and ensemble modelling/emulation. However, this introduces some potential sources of systematic errors: 1) Atmospheric CO2 concentration can misrepresent TRF under climate scenarios where the trend of CO2 emission is decoupled with the trends of other GHG emissions; 2) The assumption of time-invariant local relationship between global mean temperature/TRF and local climate variables breaks down under overshoot scenarios and over locations with strong changes in local forcing (e.g. aerosol) and energy balance (e.g. albedo feedback, land use and land cover change) (Giani et al., 2024). While the influence of pollutant emissions on PM2.5 under these scenarios can still be properly represented by our statistical model, the result from our emulator should be interpreted more cautiously under these types of climate scenarios. More advanced methods to parameterize climate effects (e.g. using cumulative GHG emissions, combining information from climate emulators) could be further explored in future work.

2. In the Gas Ratio subsection of 3.3.1 it is unclear how Gas Ratio indicates level of linearity vs non-linearity until the end of the subsection. It could help clarify the point to address this towards the beginning of the subsection.

Response: We thank the reviewer for this suggestion. We now introduce the concept of gas ratio before fig. 13

L 741 – 746: To further understand the utility of non-linear emulation, we analyse the shifts in the chemical regime of secondary inorganic aerosol formation calculating the Gas Ratio (GR) (Paulot and Jacob, 2014) over China at baseline year and under all 4 scenarios (Fig. 13):

$$GR = \frac{[NH_3] + [NH_4^+] - 2[SO_4^{2-}]}{[HNO_3] + [NO_3^+]} (7)$$

GR < 0 indicates that secondary inorganic PM2.5 is weakly sensitive to NH3 emissions through adding NH4+ to existing SO42-...

3. Many of the figures (e.g., Fig. 1, 4, 5) do not have very descriptive captions, making them rather difficult to understand. Consistent detail in the captions (e.g., Fig 6 can be easily understood as a fully standalone figure and caption) would improve the manuscript.

Response: We thank the reviewer for this suggestion. We update the captions of our figures as follows:

L 487 – 489: Fig. 3 Schematic of the 10-fold random subsampling cross-validation procedure. At each "fold", 80% of the samples (96 runs) are used to train the emulator to predict the result from

the other 20% of the samples (24 runs). The prediction is then evaluated against the GCHP-CAM output  $\Delta PM_{2.5}$  for those 24 runs.

L 522 – 524: Fig. 6 Total and regional air pollutant emissions for the four IGSM-GAINS-TAPS scenarios. Each row represents a climate scenario (Current Trend, CT; and Accelerated Actions, AA), and the emissions at 2014, and 2050 under Current LEgislation (CLE) and Maximum Feasible Reduction (MFR) air pollution control scenarios are shown for each species.

L 640 – 641: Fig. 9 Spatial patterns of Sobol Total Sensitivity Indices (0-1) for each predictor for  $\Delta PM_{2.5}$ . The indices indicate the fraction of output variance attributable to each input variables at each grid cell.

L 718 – 719: Fig. 12 GPR and MLR emulator errors relative to GCHP-CAM simulated  $\Delta PM_{2.5}$  over the 4 IGSM-GAINS-TAPS scenarios at 2050 (relative to 2014). Both emulators use the same geographic weighting scheme for pollutant emissions.

L 790 – 792: Fig. 15 Changes in absolute error (relative to GCHP-CAM output  $\Delta PM_{2.5}$ ) when no dispersion kernel (geographic weighting of pollutant emissions) is implemented. Red (positive) indicates that turning off dispersion worsens the performance (increasing error), blue (negative) indicates the opposite.

L 801 – 803: Fig. 16 Changes in absolute error (relative to GCHP-CAM output  $\Delta PM_{2.5}$ ) when the dispersion length scales ( $L_i$  in Eq. 7) for all pollutants in the geographic weighing scheme are is halved. Red (positive) indicates that turning off dispersion worsens the performance (increasing error), blue (negative) indicates the opposite.

L 827 – 829: Fig. 17 Changes in absolute error (relative to GCHP-CAM output  $\Delta PM_{2.5}$ ) when the dispersion length scales ( $L_i$  in Eq. 7) for all pollutants in the geographic weighing scheme are is doubled. Red (positive) indicates that turning off dispersion worsens the performance (increasing error), blue (negative) indicates the opposite.

**Minor comments:**

- 1. Use colorblind friendly colors for figures 3, 4, 8, 10.
- 2. Use consistent regional colors between figures 3 & 4.

Response: We thank the reviewer for these suggestions. We tried to reducing the number of colors and create distinctions by hatching, but the resulting figure is confusing. While it is challenging to provide a completely colorblind-friendly palette that has 18 (number of IGSM regions) distinct colors, we believe using a consistent and more colorblind-friendly palette ("tab20") between figures 3 and 4 would still improve the readability. We also switch to more colorblind-friendly palettes for figures 8 (viridis) and 10 (Okabe-Ito):

Revised figure 10

3. Line 134: Iman-Conover Transform is not described well enough for a reader to understand what it is.

Response: We thank the reviewer for pointing out this ambiguity. We rewrite that section to provide clearer explanation of how we apply the I-C transform procedure:

L 316 – 322: ...control policies. We then use the Iman-Conover Transform (Conover and Iman, 1982) to impose the correlation matrix for the independent and uncorrelated pairs of LHS scaling factors. The Iman-Conover Transform first transforms the sample pairs to an approximately multivariate normal distribution. Then Cholesky decompositions are used to impose the correlation matrix to the distribution, resulting in a matrix that can be applied to rearrange the sample pairs by ranking. This results in correlated pairs of scaling factors that allow us to focus on sampling the more probable parts of the input space (due to co-emissions), while preserving the marginal distributions of individual variables (i.e. uniform distribution over their respective ranges).

4. Figure 11: Remove uncoupled right parentheses ")" from colorbar label.

Response: We thank the reviewer for spotting this error. We replot fig. 11 accordingly:

**Difference in $\Delta PM_{2.5}$ with GCHP-CAM output**

Replotted figure 11

- 5. Line 198: Change "points" to "point".
- 6. Line 228: Change "mode" to "made.
- 7. Line 231: Remove "to" at the end of the line

Response: We thank for reviewer for point out these errors. Revised as suggested

8. Line 333: Range of results within 2 standard deviations are presented in confusing way (potential fix "(82.9%)-(96%)"

Response: We thank for reviewer for point out this ambiguity. We revised the sentence as:

L 683 – 685: ...Generally, the emulator performs comparably to that in the random subsampling evaluation ( $R^2 = 0.94 - 0.99$ , MAE =  $0.20 - 0.42 \mu g m^{-3}$ ). 58.7 – 84.8% and 82.9% (96%) 58.7% (82.9%) – 84.8% (96%) of the grid cells have...

- 9. Line 345: Change "agrees" to "agree"
- 10. Line 360: Change "expect" to "except"
- 11. Line 428: Remove "of"

Response: We thank for reviewer for point out these errors. Revised as suggested

[revised manuscript text omitted]